# Influenza A virus undergoes compartmentalized replication in vivo dominated by stochastic bottlenecks

Katherine A. Amato[1], Luis A. Haddock III [2], Katarina M. Braun [2], Victoria Meliopoulos [3], Brandi Livingston[3], Rebekah Honce [3], Grace A. Schaack[1], Emma Boehm[2], Christina A. Higgins [1], Gabrielle L. Barry[2], Katia Koelle [4], Stacey Schultz-Cherry [3], Thomas C. Friedrich [2,5] & Andrew Mehle [1]✉

Transmission of influenza A viruses (IAV) between hosts is subject to numerous physical and biological barriers that impose genetic bottlenecks, constraining viral diversity and adaptation. The bottlenecks within hosts and their potential impacts on evolutionary pathways taken during infection are poorly understood. To address this, we created highly diverse IAV libraries bearing molecular barcodes on two gene segments, enabling high-resolution tracking and quantification of unique virus lineages within hosts. Here we show that IAV infection in lungs is characterized by multiple within-host bottlenecks that result in "islands" of infection in lung lobes, each with genetically distinct populations. We perform site-specific inoculation of barcoded IAV in the upper respiratory tract of ferrets and track viral diversity as infection spreads to the trachea and lungs. We detect extensive compartmentalization of discrete populations within lung lobes. Bottleneck events and localized replication stochastically sample individual viruses from the upper respiratory tract or the trachea that become the dominant genotype in a particular lobe. These populations are shaped strongly by founder effects, with limited evidence for positive selection. The segregated sites of replication highlight the jackpot-style events that contribute to within-host influenza virus evolution and may account for low rates of intrahost adaptation.

---

[1] Department of Medical Microbiology & Immunology, University of Wisconsin-Madison, Madison, WI 53706, USA. [2] Department of Pathobiological Sciences, University of Wisconsin School of Veterinary Medicine, Madison, WI 53706, USA. [3] Department of Infectious Diseases, St. Jude Children's Research Hospital, Memphis, TN 38105, USA. [4] Department of Biology, Emory University, Atlanta, GA 30322, USA. [5] Wisconsin National Primate Research Center, University of Wisconsin-Madison, Madison, WI 53715, USA. ✉email: amehle@wisc.edu

The constant evolution of influenza viruses results in recurring seasonal epidemics and has the potential to initiate pandemics in the human population. Viral evolution occurs on multiple scales. Influenza A virus evolves rapidly on the global scale, where population-level immunity positively selects new antigenic variants, necessitating frequent reformulation of the seasonal influenza vaccine[1–3]. Yet, on smaller scales, variants with a predicted fitness advantage rarely rise to high frequency within an acutely infected host or transmit to a recipient, even in the face of vaccine-induced immunity[4–7]. Within-host variation is low and genetic drift plays a large role[8]. The level at which selection acts is currently unclear. To reconcile these observations, it is currently thought that positive selection drives more deterministic processes at the global scale while genetic drift dominates within hosts and during local transmission[9].

Highly pathogenic H5N1 avian influenza viruses repeatedly spill over into humans and this process can be used as an example to highlight the importance of evolutionary constraints. As few as five amino acid changes are sufficient to adapt these viruses for transmission in mammals, with some of these variants already circulating in avian hosts[10–12]. Despite this relatively modest genetic barrier, there has been no sustained person-to-person transmission. This suggests additional evolutionary processes may raise the barrier to human adaptation by reducing the likelihood that the correct combination of mutations can arise in the correct context or without deleterious mutations over the course of a typical acute infection. Influenza virus infection occurs in heterogeneous cell populations within complex anatomical structures throughout the respiratory tract[13]. Initial sites of infection are influenced by the tissue-specific distribution of sialic acid receptors and their topology. Avian-origin viruses like highly pathogenic H5N1 strains utilize α2,3-sialic acids for attachment and entry, which are enriched in the lower respiratory tract[14–16]. However, recent evidence shows that viruses in the upper respiratory tract, specifically those replicating in the soft palate or nasal epithelial cells, are the ones that contribute most to the population that is transmitted in animal models[17–19]. Therefore, viruses spilling over from avian hosts must not only acquire mammalian-specific adaptive changes but also migrate within the host to sites of transmission prior to initiating person-to-person spread.

Anatomical structures within the respiratory tract may enhance viral evolution and replication by creating local sites with a high multiplicity of infections that increase complementation[20]. Conversely, these same structures may also restrain evolution by creating distinct compartments of replication that are physically separated, precluding reassortment[21]. How viral movement in the respiratory tract and potential compartmentalization affect within-host evolution is poorly understood. A clearly defined population structure is required to accurately model and predict influenza virus intrahost evolution and to elucidate how within-host processes link with evolution on the global scale. Although error-prone genome replication generates influenza A viruses with distinct genotypes, the relatively low levels of naturally occurring variation within hosts do not provide sufficient information for fine-grained analysis of population dynamics.

In this work, we overcome this limitation by introducing a neutral barcode of 10 random nucleotides into two segments of the influenza virus genome and creating rich viral populations with $\sim 0.6$–$3 \times 10^5$ uniquely quantifiable members. Using these barcoded viral populations, we capture soft selective sweeps in cell culture and show adaptive changes arose independently multiple times, yet only one lineage became dominant. Infection in ferrets reveals a high degree of compartmentalization, as the virus migrated from the upper respiratory tract to the lung. Bottlenecks between sites coupled with heterogeneous replication success in

the lobes of the lung lead to stochastic sampling of individual viruses from the upper respiratory tract or the trachea that became the dominant lineage in lung lobes, while there is no evidence of positive selection. Thus, viruses infecting the lung do not constitute a large homogeneously mixed population, but rather multiple isolated populations that each undergo bottlenecking events that can severely constrain population diversity and the potential for selection of fit variants.

## Results

**Large-scale incorporation of barcodes avoids artificial bottlenecking of variants.** Naturally occurring genetic diversity in influenza virus populations is poorly suited for high-resolution characterization of population-level dynamics and cannot accurately enumerate the full spectrum of individual virions present. Many virions will not have genetically unique markers and genetic mutations may affect the fitness of the virus, biasing the representation of specific variants in the population and precluding their use as a neutral marker. To better resolve and quantify the dynamics of the IAV population, we introduced dual molecular barcodes into the genome of the influenza virus isolate A/California/07/2009 (H1N1; CA07) to create individual viruses that are uniquely recognizable and quantifiable via deep sequencing (Fig. 1a). Barcodes of 10 randomized nucleotides were introduced onto the *HA* and *PA* segments that were subsequently used for high-efficiency virus rescue (Fig. 1b). HA is under selective pressure as the viral attachment protein and the principal target of neutralizing antibodies. We might therefore expect barcodes embedded in *HA* to increase or decrease in frequency as a result of selection on *HA* that allows hitchhiking of the linked barcode. Conversely, PA is subject to less intense selection pressure, and barcodes on this segment may be expected to better represent the total population size. For both *HA* and *PA*, barcodes were encoded between the ORF and the UTR. Packaging and bundling signals were duplicated downstream of the barcode to ensure proper gene replication and virion formation[22–24]. This strategy parallels prior work where small-scale barcoded libraries were successfully used to quantify transmission bottlenecks[17]. The utility of the system was further increased by using the PA reporter construct that co-transcriptionally expresses Nanoluciferase (PASTN)[22]. Finally, the *HA* segment encodes an additional "registration mark," six nucleotides creating either NheI or PstI restriction sites, that allows us to index and identify separate libraries of *HA* variants.

The limited efficiency of virus rescue can introduce artificial bottlenecks[25]. To increase efficiency and recover a larger, more diverse barcode population, we reduced the number of individual plasmids needed for virus rescue[26]. We combined the 6 non-barcoded segments of CA07 IAV onto a single plasmid, reducing the entire reverse genetics system from 8 to 3 plasmids (Fig. 1c). Rescue titers of the 3-plasmid system increased >200-fold compared to the 8-plasmid system. Titers were further increased by co-expressing transmembrane protease serine 2 (TMPRSS2) during virus rescue, which cleaves and activates HA. We performed 120 parallel virus rescues, pooled the resultant supernatants, and passaged them at high representation to ensure unbiased barcode distribution (Fig. 1b). NheI and PstI registration marked libraries were prepared independently.

**Viral barcodes reveal selective sweeps in vitro.** Deep sequencing revealed ~180,000 unique *PA* barcodes and over 238,000 *HA* barcodes in each of our original virus libraries (Fig. 2a). Given that each library has an invariant registration mark, we could measure the fidelity of the quantification pipeline by assessing the number of reads that do not match the predicted registration

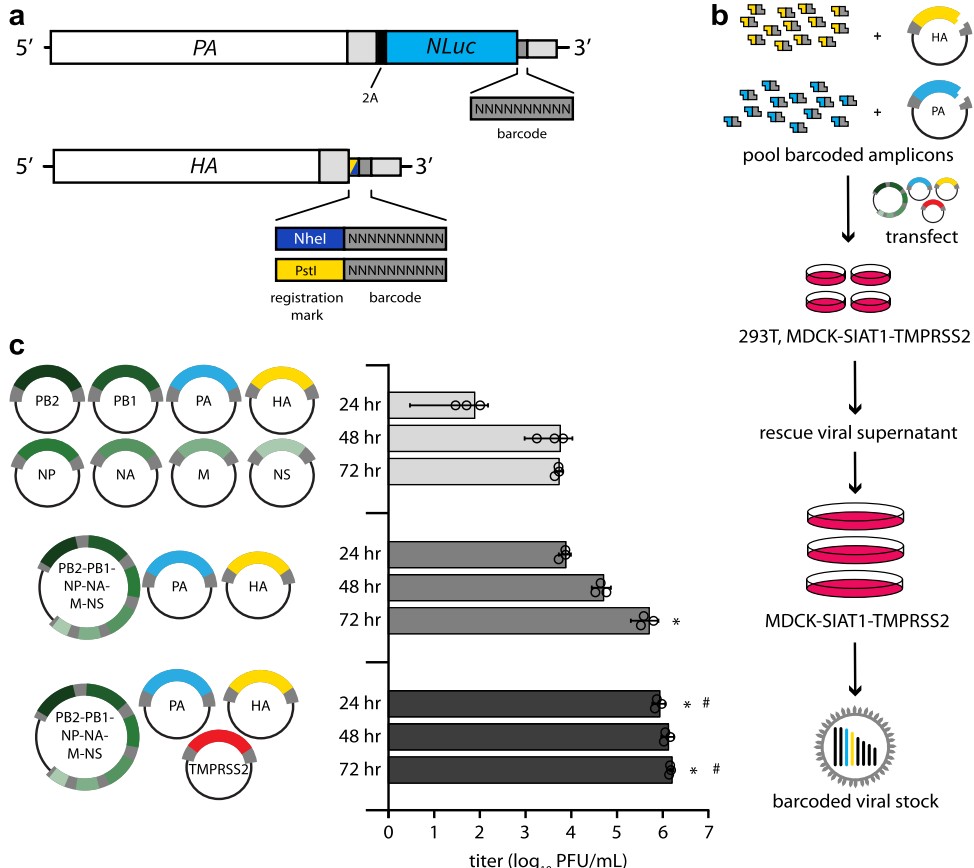

**Fig. 1 Creation of molecularly barcoded influenza A virus populations. a** Molecular barcodes containing 10 randomized nucleotides were encoded downstream of the open reading frame in the *PA* and *HA* genes, shown as cRNAs. A registration mark was also added to *HA* to distinguish unique barcode libraries. Sequences were repeated downstream of the barcode to maintain contiguous packaging signals required for replication, and silent mutations were introduced into the open reading frame to avoid direct repeats. **b** Experimental overview where randomized barcodes were cloned into reverse genetics vectors followed by large-scale, parallel virus rescues to ensure unbiased barcode distribution. **c** Optimized rescue plasmids enhance viral yield. Rescue efficiency was determined by measuring viral titers at the indicated times post-transfection with the standard 8-plasmid system, a consolidated 3-plasmid system, or the 3-plasmid system plus a vector expressing TMPRSS2. (data presented as the mean of $n = 3 \pm$ sd. ANOVA with Tukey's post hoc, $*p < 0.05$ relative to 8-plasmid rescue, $\#p < 0.05$ relative to three-plasmid system.) Source data are provided as a Source Data file.

mark sequence. Over 99% of mapped reads perfectly matched the appropriate registration marks on *HA*, with the majority of those that did not match differing by a single nucleotide from the intended registration mark, indicating the neutrality of the registration mark sequence and overall high fidelity of our sequencing pipeline. Sequencing errors may create the appearance of unique barcodes, artificially inflating our diversity metrics. To minimize this possibility, we performed clustering on aligned sequences that created networks connecting a parental barcode with mutational derivatives[27]. All reads in the cluster were then assigned to the parental barcode before quantitation.

Viral populations were characterized by: richness, the total number of unique lineages present; Shannon's diversity and the Gini-Simpson index, metrics that consider both the presence and relative abundance of a lineage; and, evenness, a parameter that compares the frequencies of all lineages in the population to that of a theoretically evenly distributed population[28–30]. Shannon's diversity is more influenced by richness where very rare lineages have large effects, whereas the Gini-Simpson gives more weight to dominant lineages and is thus influenced more by evenness[31]. To account for the possibility that the underlying structure and size of our populations may bias their diversity assessments, we report both. Barcode enumeration for viruses recovered from the transfected cell supernatant (i.e., passage 0, P0) revealed rich and highly diverse populations with evenly distributed lineages

(Fig. 2a, b). However, a single passage at an MOI ~0.01 with a total of ~$10^6$ virions to create amplified P1 stocks resulted in highly skewed populations in which individual lineages dominated the population, evidenced by dramatic drops in Shannon's diversity and evenness (Fig. 2b). The dominant barcodes in each library represented 64–77% of the population. Neither of these barcodes was dramatically over-represented in the plasmid or P0 stocks. In fact, the most abundant lineages in the P0 rescue stocks did not become the most prevalent in our amplified stocks.

Changes in population composition can occur through drift or selection. We passaged the virus at a large effective population size, minimizing the propensity for drift and raising the possibility that the expanding lineages became dominant because they had acquired a selective advantage. Whole-genome sequencing of the P1 stocks identified the single nucleotide variants (SNVs) A540G (numbering based on cRNA) in almost 50% of reads from NheI registration mark viruses and A543G in almost 70% of reads from PstI marked viruses (Fig. 2c), mirroring the abundance of the dominant barcode. These mutations code for HA K153E and K154E (H1 numbering), respectively. Importantly, these changes in HA had previously been identified as adaptations that provide a growth advantage to CA07 in cell culture[32]. Long-read sequencing showed that the dominant barcode was linked to the adaptive mutations (Fig. 2d, left). 62–75% of reads containing the dominant barcode also encoded

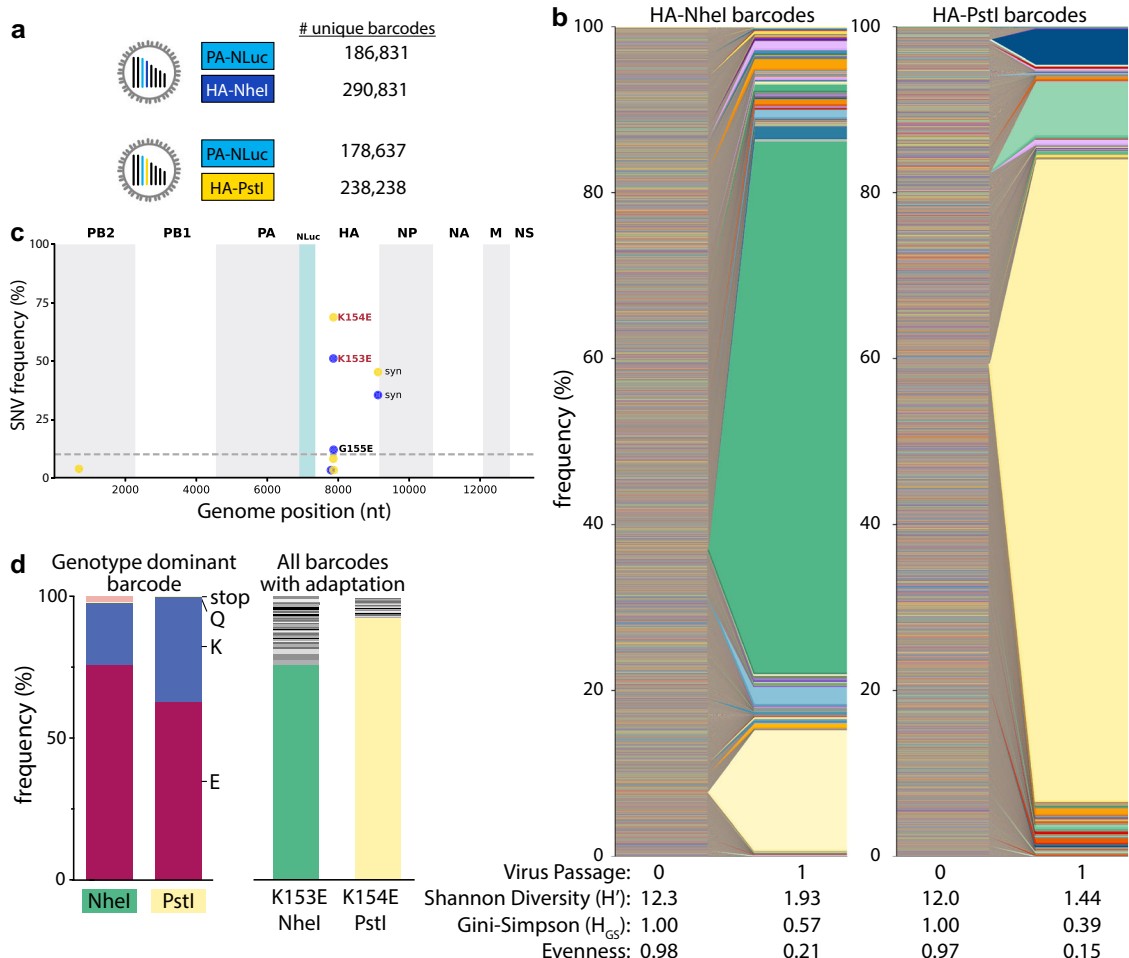

**Fig. 2 Single viruses seed selective sweeps in *HA*. a** Creation of two dual-barcode libraries with distinct registration marks and uniquely addressable members. **b** Frequency of each lineage as a fraction of total population size. Colors indicate unique barcode identities. **c** Whole-genome sequencing identifies adaptive variants in *HA*. Individual single nucleotide variant (SNV) frequencies are indicated at each nucleotide position in a concatenated IAV genome for each library. **d** Long-read sequencing reveals selective sweeps by linking adaptive mutants in HA to single dominant barcodes. The frequencies of mutations coding for the indicated change that is linked to the dominant barcode are shown for both libraries (left). The actual frequency of K > Q and K > stop mutants versus the contribution from long-read sequencing errors to their appearance is unknown. The frequencies of all barcodes linked to the adaptive glutamic acid variant are indicated (right), with the dominant barcode in each library colored as in **b**. Source data are provided as a Source Data file.

the adaptive mutation. Similarly, the adaptive mutation was primarily linked to the dominant barcode (Fig. 2d, right). However, up to 24% of reads encoding these adaptive variants were associated with different very-low-frequency viral lineages. These data are consistent with a soft selective sweep where *HA* A540G or A543G arose on multiple genetic backgrounds, even though only one lineage ultimately became the most abundant, possibly suggesting clonal interference. These observations demonstrate that our barcoded viruses capture lineage dynamics and selective processes at extremely high resolution.

**Pre-adaptation creates large and diverse viral libraries**. To create diverse libraries without tissue-culture-induced skewing, we made new libraries on a "pre-adapted" HA K153E background. The libraries contained at least 57,000 unique members with high diversity and evenness (Fig. 3a, b). Multicycle growth curves of HA K153E NheI- or PstI-marked barcoded viruses were indistinguishable (Fig. 3c). Insertion of the barcode cassette and the *PA* reporter gene slightly reduces titers when compared to HA K153E alone, consistent with prior results[22,33]. The vast majority of lineages were present at low frequency, and no single lineage in

our amplified stocks was present at a frequency above 0.6% for HA, or above 1.2% for PA barcodes (Fig. 3d). We utilized the HA-NheI stock for extensive quality control of our barcode enumeration pipeline. Libraries were prepared from four independent RNA extractions and sequenced. Lineage frequencies from the four replicates showed a strong correlation (Supp. Fig. 1A). There was a high degree of overlap between replicate runs, with sequences present in all four replicates accounting for over 93% of all read counts (Supp. Fig. 1B). Yet, when read counts for identical sequences were collapsed to their corresponding barcode, only ~37% of barcodes were identified in all replicate runs (Supp. Fig. 1C). This apparent discrepancy is due to variable detection of extremely low-frequency lineages. The frequency of an individual lineage was correlated with the number of replicates in which it was detected (Supp. Fig. 1D). Rarefaction-extrapolation curves for each replicate overlapped, indicating reproducible sampling (Supp. Fig. 1E). However, they also suggest that our sampling has not reached saturation and that some low-frequency lineages may not be detected. Further, when compared to the simulated sampling of a population with evenly distributed barcodes, our replicates diverged from the ideal population. This is consistent with the presence of extremely low-frequency

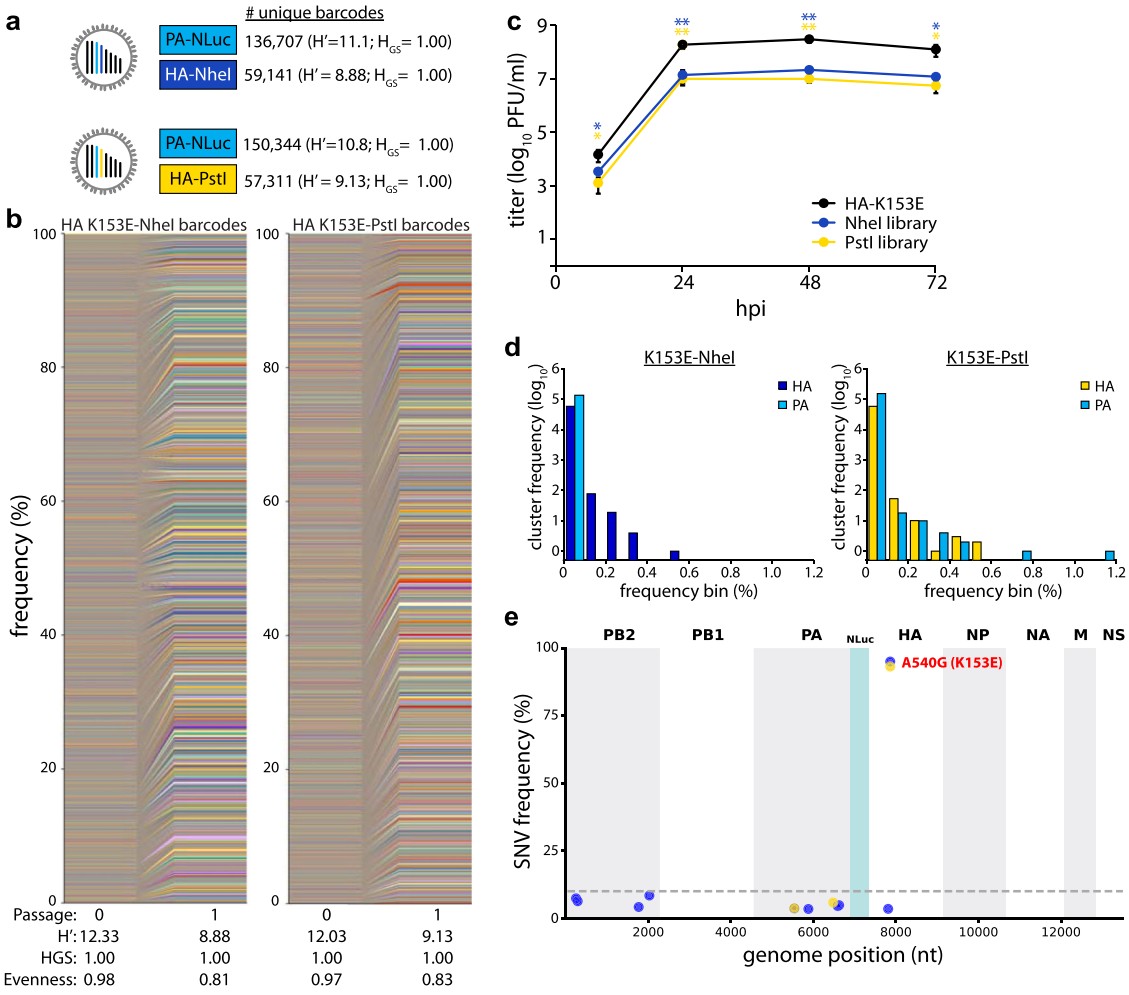

**Fig. 3 Generation of large and evenly distributed dual-barcoded virus libraries. a, b** Properties of the A/CA/07/2009 HA-K153E PASTN virus libraries. Data are from a single sequencing replicate; see Supplemental Figs. 1, 2 for additional analyses. **c** Multistep growth curves of dual-barcoded PASTN libraries compared to the parental strain A/CA/07/2009 HA-K153E. Viral titer was measured by plaque assay (mean of $n = 3 \pm$ sd, two-way ANOVA with Tukey's post hoc, $*p < 0.05$ and $**p < 0.01$ compared to the parental). **d** Frequency distribution of unique barcodes on *HA* and *PA* binned in 0.1% increments. **e** Whole-genome sequencing was performed on amplified viral stocks and SNV frequencies are indicated at each nucleotide position for each library. Source data are provided as a Source Data file.

lineages. These data highlight the complexity of our viral populations, our ability to reliably detect individual members, and the inherent limitations of this approach. Replicate sequencing of the *PA* barcode and the HA-PstI stock produced similarly well-correlated results (Supp. Figs. 1F, 2A–C). No additional SNVs were detected at high frequency in our libraries, while K153E remained fixed (Fig. 3e).

**Rich and diverse populations replicate in mice.** The dual barcoded virus libraries provide a key opportunity to quantify population dynamics in vivo. Mice are frequent models for influenza virus replication, pathogenesis, and immune response[34]. Mice were inoculated with the CA07 PASTN-barcoded virus libraries containing either barcoded HA-K153E populations or a non-barcoded HA-K153E control. Weight loss was similar for all conditions (Fig. 4a), and viral titers in the lungs were not significantly different at either 3 or 6 dpi (Fig. 4b). Thus, the introduction of barcodes onto *HA* does not compromise replication in vivo.

Deep sequencing of mouse lung homogenates revealed that the majority of mice harbored diverse viral lineages (Fig. 4c, d). Mouse infections were characterized by a high richness, where

approximately one-third of lineages present in the stock were detected in the lungs at 3 dpi. The frequency of lineages in the NheI-marked stock was moderately predictive of their abundance in the mouse at 3 dpi (Pearson's $R > 0.57$) (Supp. Fig. 3A). For example, the most abundant lineage in the HA-NheI stock was also the most abundant lineage in two mice at 3 dpi (Fig. 4c, Supp. Fig. 3A). However, there are notable exceptions in which low-frequency lineages in the stock rose to high relative abundance in the mouse (Supp. Fig. 3A). In addition, we cannot exclude the possibility that some of our barcodes are derived from residual viruses from the inoculum that did not initiate a productive infection. Mice infected with PstI-marked libraries showed less correlation between the inoculum and lungs at 3dpi (Supp. Fig. 3A). For all viral libraries, the titers decreased from 3 to 6 dpi, as did richness and overall diversity (Fig. 4c, d). In an extreme example, a single lineage in mouse 11 rose to over 40% prevalence at 6 dpi (Fig. 4C, Supp. Fig. 3B). The correlation between lineage frequency in the stock and the mouse lung was greatly diminished at 6 dpi (Supp. Fig. 3). The lineage identities are highly heterogeneous among mice, with a small fraction shared across animals, suggesting that barcodes themselves are not under selection in vivo (Fig. 4e). Together, these data show virus populations replicating in mice approximate the diversity present

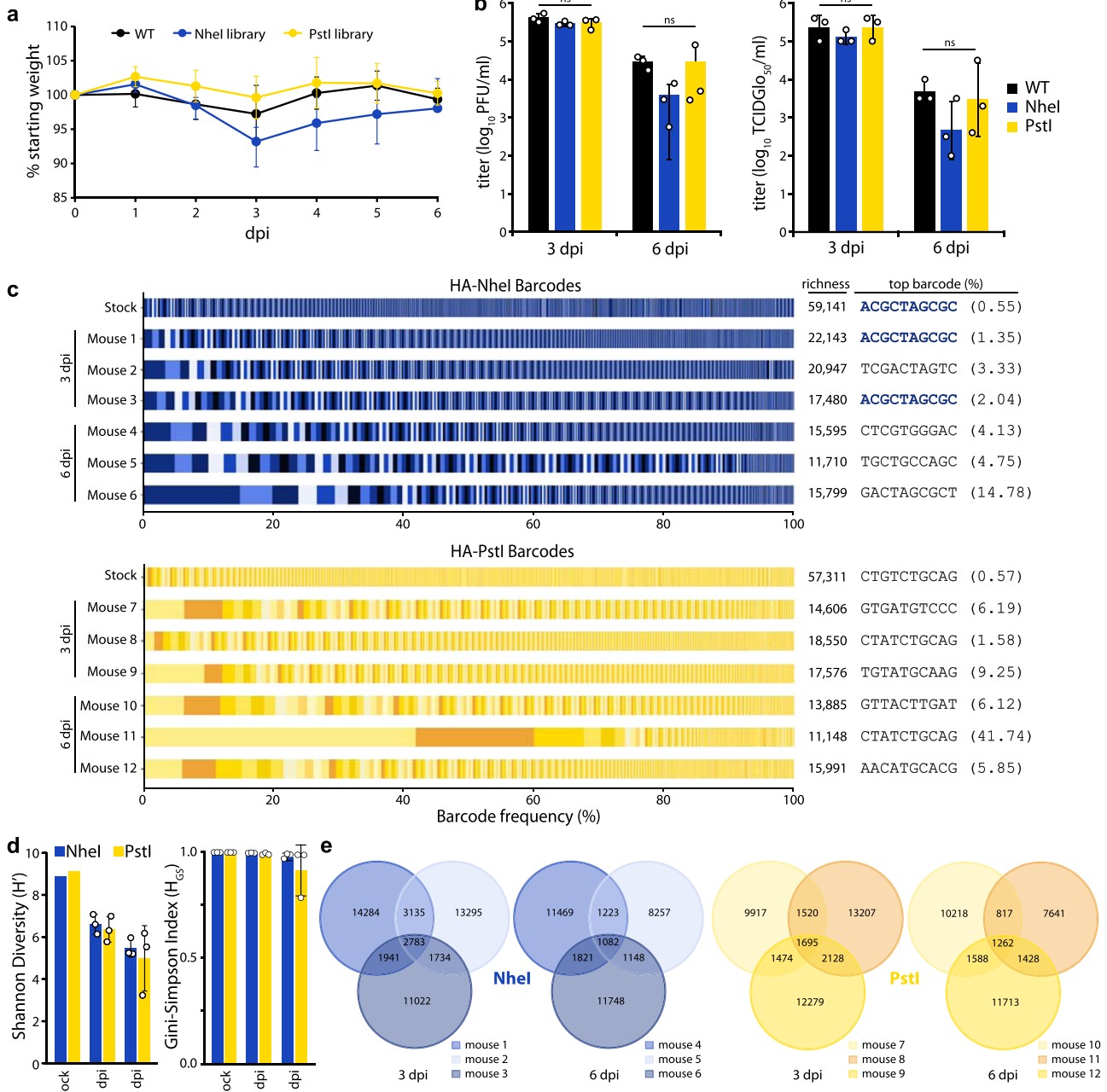

**Fig. 4 Replication of diverse virus populations in mice. a** Mice were inoculated with 105 TCIDGlo50 of virus-containing barcoded PASTN with either HA-K153E or barcoded variants and body weights were measured daily. Half of the mice were killed at 3 dpi. Data presented as mean ± sd for $n = 6$ 1–3 dpi, and $n = 3$ for 4–6 dpi. **b** Viral titers in mouse lungs harvested at 3 and 6 dpi were determined by plaque assay (left) or TCIDGlo50/mL (right). Data presented as the mean of $n = 3 \pm$ sd and analyzed by one-way ANOVA. n.s. = not significant. **c** Barcodes in the viral stock and mouse lungs were quantified and the frequency of clustered HA barcodes as a fraction of total population size is indicated. Each color in a series represents an individual barcode cluster. A total number of unique barcode clusters per sample and the most abundant barcode with its frequency is listed on right. **d** Shannon's diversity index (left) and Gini–Simpson index (right) for viral populations in the stock and mouse lungs (mean of $n = 3 \pm$ sd). **e** Venn diagrams displaying the number of unique and shared lineages within each mouse for NheI and PstI libraries. Source data are provided as a Source Data file.

in the inoculum, as might be expected from a high-dose challenge. Within-host richness decreases as the infection are resolved, with lineages lost as the overall population size decreases.

**Seeding and localized replication dynamics result in the compartmentalization of viral lineages in the ferret lower respiratory tract.** Ferrets are often considered the "gold standard" infection model with lung physiology, sialic acid distribution, pathogenesis, and transmission capacity that are all similar to

humans[35,36]. We intranasally inoculated 3 ferrets with the HA-K153E dual barcoded library containing the NheI registration mark. We used a site-specific inoculation strategy in which the inoculum is retained in the upper respiratory tract without unintentional introduction into the trachea or lower respiratory tract[37], allowing us to track the natural movement of the virus. Ferrets were monitored daily for signs of infection with nasal washes obtained 1, 3, and 5 dpi. Ferrets exhibited slight weight loss over the course of infection (Fig. 5a), consistent with prior

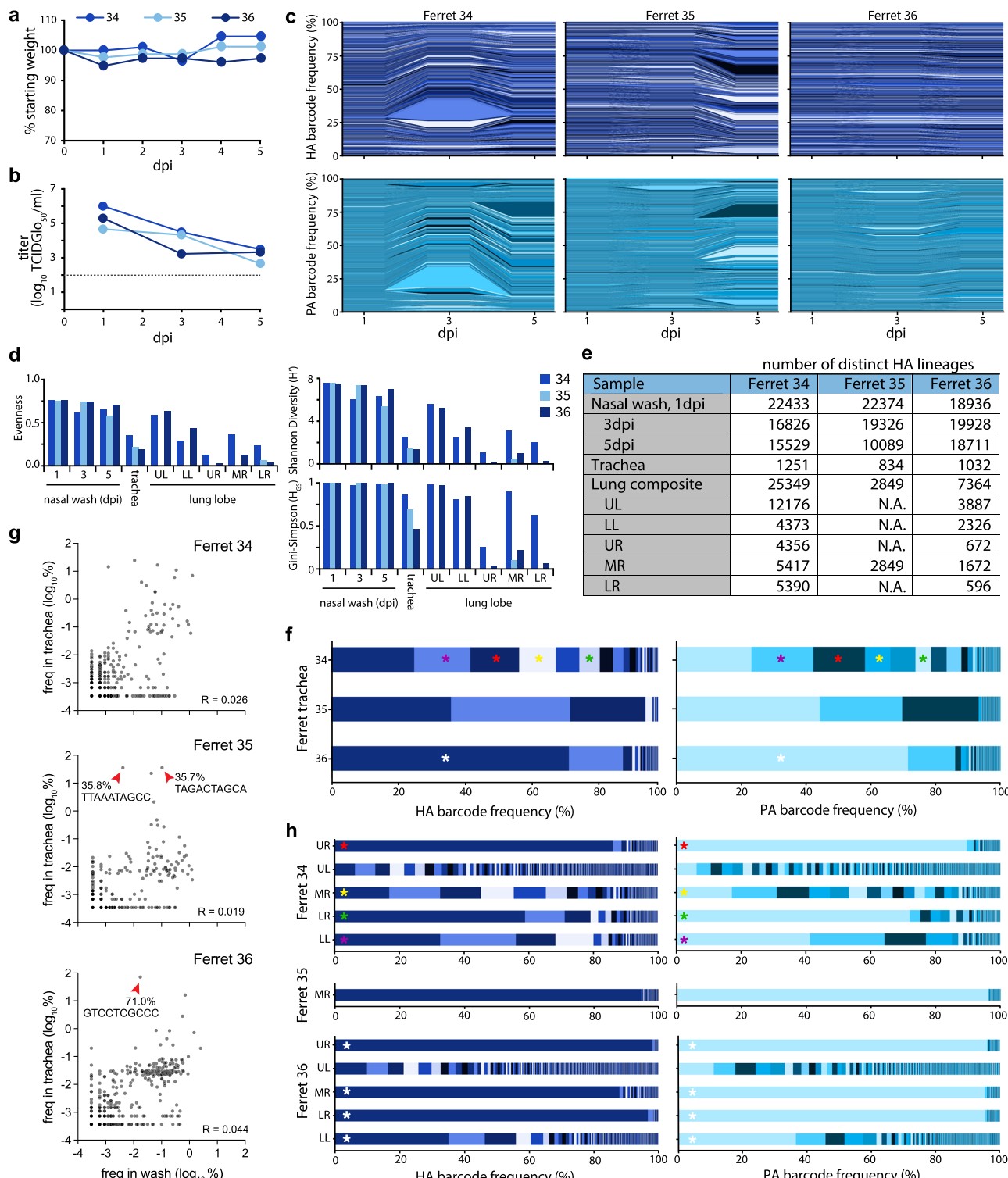

**Fig. 5 Population diversity is reduced when the influenza virus moves from the upper to the lower respiratory tract in ferrets. a** Ferrets were inoculated with a site-specific intranasal dose of $10^5$ PFU of dual-barcoded A/CA/07/2009 HA-K153E PASTN virus containing the NheI registration mark. Ferret weight was monitored daily. **b** Viral titer in ferret nasal washes were determined by TCIDGlo50. **c** Changes in frequency for HA (top) and PA (bottom) barcodes present in nasal wash samples over the course of infection. **d** Evenness, Shannon's diversity index, and Gini-Simpson index for the indicated viral populations. **e** The richness of each tissue is indicated by the number of distinct barcodes in each sample. N.A. = not attempted. **f** Virus was recovered from the trachea at 5 dpi and the frequency of barcodes was determined. Distinct dominant barcodes are identified by colored asterisks. **g** HA barcode frequencies were compared between the nasal wash (3 dpi) and trachea (5 dpi) for ferret 34, 35, and 36. Red arrowheads highlight dominant barcodes with frequency > 30%. R = Pearson's correlation coefficient. Note that barcodes unique to the nasal wash or trachea are not plotted here. **h** Virus was recovered from lung lobes 5 dpi and the frequency of barcodes was determined. Lung lobes: upper left (UL), lower left (LL), upper right (UR), middle right (MR), lower right (LR). Distinct dominant barcodes are identified by colored asterisks, matching those in **g** where appropriate. The limit of detection for viral titer assays is indicated by a dashed line in **b**. Source data are provided as a Source Data file.

work[33]. Similarly, we detected high viral titers in nasal washes 1 dpi that declined over time (Fig. 5b).

Sequencing revealed rich and complex viral populations in the nasal washes of ferrets, consistent with a high dose inoculation in this compartment (Fig. 5c–e). Frequency trajectory plots showed heterogeneous and well-mixed viral populations undergoing little if any, selection or bottlenecks in the upper respiratory tract (Fig. 5c). This is consistent with a highly diverse population with generally even lineage distribution, reflected by high Shannon's diversity and Gini-Simpson indices and evenness (Fig. 5d). Approximately 19,000–22,400 unique lineages were detected in each nasal wash at 1 dpi, and richness remained high over time, with at least 16,800 lineages at 3 dpi and 10,000 lineages at 5 dpi (Fig. 5e). Sequencing of the barcode on *PA* revealed remarkably similar lineage dynamics (Fig. 5c). The observation that frequencies of *HA* and *PA* barcodes moved in parallel is perhaps surprising given that these are unlinked genes and that influenza viruses can undergo frequent reassortment[38].

Intranasally inoculated viruses spread throughout the respiratory tract by 5 dpi. Low levels of infectious virus were present in the trachea of ferrets 35 and 36 (Supp. Fig. 4A), whereas deep sequencing detected IAV genetic material in all trachea samples (Fig. 5f, g). Compared to the nasal wash, population richness dropped significantly, with fewer than 1300 lineages present in any of the trachea samples. Lineage distribution differed in the trachea of each animal, ranging from a more diverse population in ferret 34 to a largely homogenous population in ferret 36, in which a single lineage accounted for 71% of the population (Fig. 5f). Because our site-specific inoculation requires virus replication in the upper respiratory tract prior to movement into the trachea or lower respiratory tract, we used lineages present in nasal washes at 3 dpi as a comparator for populations in the trachea and lungs at 5 dpi. Lineage frequency is poorly correlated between nasal washes and the trachea (Fig. 5g). Migration into the trachea is associated with a drastic reduction in richness, a poor correlation with the source, and skewed distribution of the resultant population, indicating that viral population bottlenecks between compartments and founder effects may play a role during the seeding of the trachea from the upper respiratory tract.

The virus also spread to the lungs of infected animals (Fig. 5h, Supp. Fig. 4A, B). Moderate titers were detected in all five lung lobes in ferret 34, even though infectious viral titers in the trachea were below the limit of detection for this animal. Infectious virus was also detected in all lung lobes for ferret 36, but only the middle right lobe for ferret 35. Thus, while the virus must traverse the trachea to access the lungs, the presence of the virus in the trachea 5 dpi was not predictive of the extent of spread in the lungs. Moreover, populations in lung lobes had higher lineage richness than that detected in the trachea at 5 dpi; the vast majority of lineages in the lung were not detected in the trachea (Supp. Fig. 4C). This raises the possibility that population richness in the trachea at 5 dpi shrank significantly from the populations at earlier time points that may have seeded infection in the lung, or that virus can transit through the trachea to directly inoculate the lung.

Lineage analysis revealed heterogeneous populations of barcodes in each of the distinct lobes (Fig. 5h). Over 25,000 lineages were detected across the five lobes in ferret 34. However, each lung lobe of ferret 34 had a different dominant barcode sequence, and when this same barcode was detected in other lung lobes its frequency varied. The only infected lung lobe of ferret 35 had very low viral titers and was dominated by a single lineage reaching ~95% abundance (Fig. 5h, Supp. Fig. 4B). Ferret 36 yielded another outcome, where the same lineage was dominant at a frequency of 35–98% in each lobe except for the upper left. Like ferret 34, the upper left lobe of ferret 36 maintained a richer and

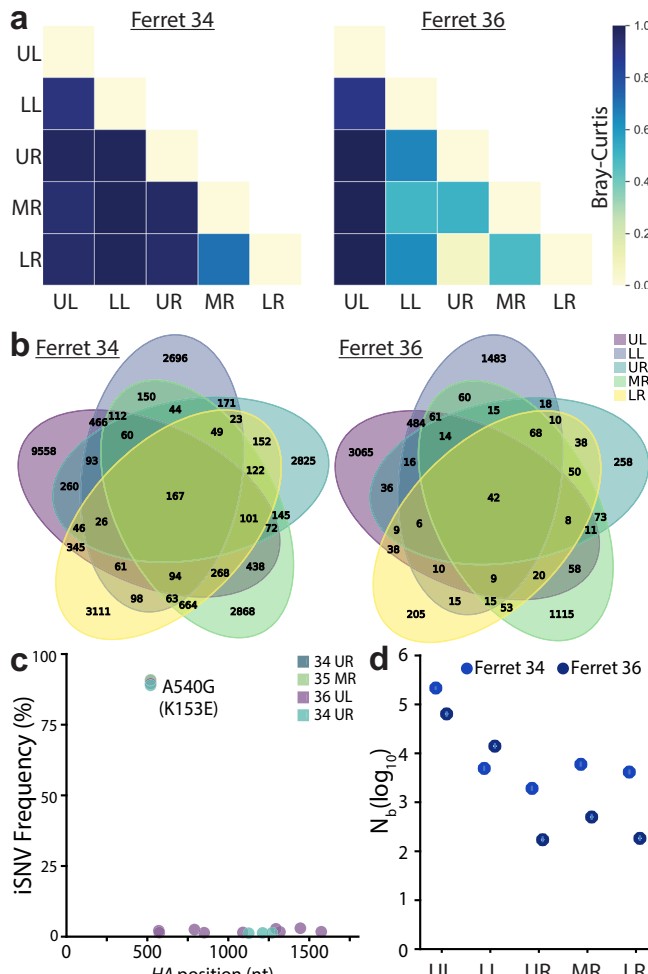

**Fig. 6 Lung lineage diversity and population bottlenecks are dominated by stochastic pressures. a** Pair-wise Bray-Curtis dissimilarity for lung lobes within single animals. **b** The number of barcode lineages common and unique to lung lobes are illustrated. **c** Whole-HA sequencing of virus in lung homogenates identified multiple iSNVs, but did not rose above the 3% threshold set for accurate estimation of iSNV frequency. **d** Transmission bottlenecks ($N_b$) from nasal wash 3 dpi to each lung lobe at 5 dpi were calculated by maximum likelihood estimation. 95% confidence intervals are shown by the lines within the dots. Source data are provided as a Source Data file.

more diverse population. For all animals, individual lobes showed reduced diversity and evenness compared to the virus population in the upper respiratory tract (Fig. 5d). In the two animals in which the virus was detected in all lobes, the upper left lobe consistently had the highest richness, diversity, and evenness. These data suggest that anatomical features associated with each lobe, such as tracheal bifurcation patterns or bronchus size[37], may affect patterns of virus establishment and replication. The differences in the composition of the populations in each lung lobe were determined. Bray-Curtis dissimilarity assessment revealed a high degree of compartmentalization, in which each lung behaved as a distinct anatomical "island" with a unique population composition (Fig. 6a). Four lobes in ferret 36 showed lower dissimilarity, as they were dominated by the same lineage, yet the non-dominant lineages still contributed unique populations to each lobe. Only 167 lineages were common to all lobes of ferret 34, and only 42 in ferret 36 (Fig. 6b).

The drastic reduction in richness and diversity between the upper respiratory tract and the lung lobes suggested populations

were subject to selection or bottlenecks. *HA* sequencing from select lung lobes for each animal revealed that no other intrahost SNVs (iSNVs) surpassed the 3% threshold set for an accurate estimate of iSNV frequency (Fig. 6c). We, therefore, assessed the possibility that viral population bottlenecks were driving the observed reduction in richness and diversity. Note that the limited number of animals inherent in ferret experiments does not provide sufficient power to exclude a contribution of positive selection. We developed a simple multinomial model to estimate bottleneck sizes ($N_b$) as the virus transmits from the nasal wash at 3 dpi to lung lobes at 5 dpi (Fig. 6d). Specifically, this model estimated $N_b$ using data on the lineages and their frequencies in the 3 dpi nasal wash and on the number of lineages observed in a focal 5 dpi lung lobe. The model yielded maximum likelihood estimates of $N_b$ of ~66,000 (ferret 36) and ~217,000 (ferret 34) virions for the upper left lobe, and lower estimates for the other lobes where maximum likelihood estimates spanned between ~200 and ~15,000 virions (Fig. 6d). These high estimates of $N_b$ are consistent with the detection of a relatively large number of lineages in each of the lung lobes, many of which are observed at only a very low frequency in the 3 dpi nasal wash samples. Furthermore, the larger bottleneck size between the nasal wash and the upper left lobe compared to that for the other lung lobes was consistent with higher levels of viral lineage diversity detected in the upper left lobe compared to the others. Nonetheless, the bottleneck size estimates still appeared unexpectedly large to us, given the drastic reductions in richness and diversity observed in Fig. 5d, e. We, therefore, forward-simulated the multinomial model under the estimated bottleneck sizes, plotting the expected frequencies of the lineages in the 5 dpi lung lobe samples, given their frequencies in the 3 dpi nasal wash samples. These simulations predicted a high degree of similarity between lineage frequencies in the viral populations sampled in the nasal wash and lung lobes, inconsistent with the more disparate lineage frequencies observed (Supp. Fig. 5A). To identify possible reasons for these discrepancies, we modified the multinomial model simulations to incorporate environmental noise; this type of noise would be expected if sites within individual lung lobes were heterogeneous, with some allowing for efficient viral replication while others being less conducive for replication. Simulations of this modified model, parameterized with a substantial amount of lobe heterogeneity (>700-fold replication difference between the 5% and 95% percentiles of the viral population), yielded correlations more similar to those observed between nasal wash and lung lobe lineage frequencies (Supp. Fig. 5B). These results indicate that even though many lineages appear to seed the lung lobes, once in the lobes, replication of viral lineages may be subject to large stochastic effects, reducing viral diversity and evenness.

Considering the lung as a whole for ferrets 34 and 36, a substantial number of lineages were shared between the nasal wash and lung (Fig. 7a). But, this appeared to be largely driven by the rich and diverse population in the upper left lobe, as this overlap was largely lost when lobes were considered individually. Many of the dominant lineages in a lung lobe were poorly represented in the nasal wash, such as that in the upper right lobe of ferret 34 and the dominant lineage shared in 4 lobes of ferret 36 (Fig. 7b, c, Supp. Fig. 6A, C). Lineage enrichment in the lung samples compared to nasal washes revealed many lineages that were unique to nasal washes or lung lobes (Supp. Fig. 6A–C). These data show that treating the lung as a whole can lead to very different and misleading population structures, highlighting the importance of assessing each lobe individually.

The same lineage dominated in four lobes of the lung for ferret 36, whereas underlying diversity and a lack of overlap in each

population still suggest each lobe is seeded by distinct inoculation events (Figs. 5h, 6b, 7d). In this animal, enrichment for the dominating lineage appeared to occur in the trachea (Figs. 5f, g, 7A). This may be a general trend, as high-frequency lineages in the trachea were often over-represented in the lobes (Supp. Fig. 7A, B). Combined, the large differences in lineage identity and frequency show that each lobe is independently seeded, with little mixing between compartments. Moreover, comparisons across compartments suggest that, while each lobe may receive a large number of viral lineages, stochastic replication dynamics within individual lobes may play a role in substantially reducing the evenness and viral genetic diversity of the seeded lineages.

## Discussion

A quantitative understanding of population dynamics is crucial for determining how evolutionary forces shape viral populations within and between hosts. Error-prone replication by the influenza virus generates genetic diversity within an infected host. However, the full extent of that diversity does not survive transmission to a new host. Transmission events between hosts involve stringent bottlenecks, with only 2–6 viral genomes founding the next infection during aerosol transmission[8,17,39]. Here we show that the influenza virus also faces multiple bottlenecks within a host as it seeds different compartments. Rich and diverse populations in the upper respiratory tract were stochastically sampled as viruses transited into the lower respiratory tract, introducing strong founder effects that skewed the resultant population. Our data suggest a scenario in which repeated within-host bottlenecks and local heterogeneous growth severely reduce diversity, richness, and evenness, resulting in distinct, compartmentalized "island populations" in each lobe of the lung.

Our results show that influenza virus evolution within a host involves multiple stochastic or noisy processes, tempering the impact of positive selection and likely creating additional barriers for host-adaption and onward transmission. This could involve multiple physical bottlenecks as the virus sequentially infects distinct tissues and stochastic replication dynamics within a focal tissue following its infection. Our modeling provides evidence that both of these processes may be at play and suggests that replication kinetics in these largely isolated populations may play a critical role in shaping patterns of viral diversity detected at a sub-organ level. While the model simulations with heterogeneous viral replication were able to recapitulate observed patterns of viral diversity and the relatively weak correlations between nasal wash and lung lobe lineage frequencies, we cannot definitively conclude that heterogeneous viral replication is the process that drives these patterns. Alternative processes not considered explicitly here may also be able to reproduce similar patterns. For example, repeated seeding from the upper respiratory tract may allow viral lineages that reach the lower respiratory tract earlier to expand before the arrival of other viral lineages. This would have a similar effect of reducing the evenness and diversity of viral populations in the lung lobes relative to those sampled in nasal washes. We would also expect a process of repeated seeding to yield weak correlations between nasal wash and lung lobe frequencies. Of course, heterogeneous viral replication within lobes and repeated seeding are also not mutually exclusive processes, and both may be at play in shaping patterns of viral diversity in the lower respiratory tract. Future work is needed to be able to experimentally assess the relative roles of these two processes, and potentially others, in driving these patterns of viral diversity.

The high degree of compartmentalization we detected in ferrets has recently been described in other animals, including swine and guinea pigs, reinforcing the importance of these spatial patterns on viral evolution in multiple systems[40]. And while we considered

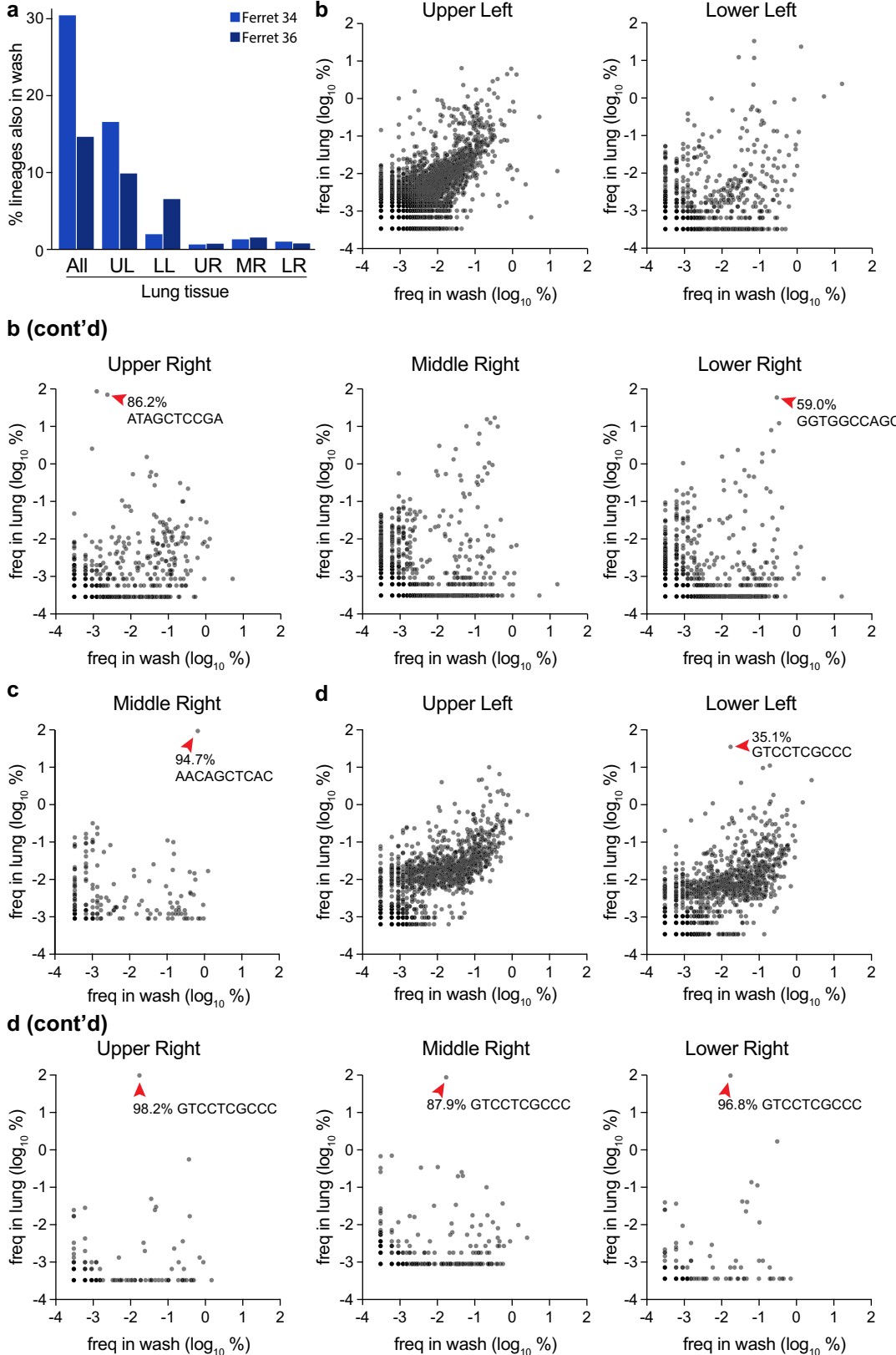

**Fig. 7 Compartmentalized infections establish replication islands within the lung. a** Overlap between barcodes in the nasal wash 3 dpi and those present in distinct lobes, or composite data for all barcodes in the lung. **b–d** The frequency of barcodes within the nasal wash is compared to frequencies within individual lobes for **b** ferret 34, **c** ferret 35, and **d** ferret 36. Red arrowheads highlight dominant barcodes with frequency >30%, which for ferret 36 is the same barcode that dominated in multiple lobes of the lung. Note that barcodes unique to the nasal wash or lung are not plotted here and can be found in Supplemental Fig. 6. Source data are provided as a Source Data file.

entire lung lobes, it remains possible that focal infections within tissues may further subdivide viral populations. In all scenarios, bottlenecks and other processes that introduce stochasticity decrease viral diversity and overall population size and are expected to increase the strength of genetic drift[41]. These processes combine to constrain diversity and evolution during influenza virus infection, which slows the fixation of adaptive variants within a host. Nonetheless, random processes like those driven by bottleneck events can also be advantageous in certain situations, creating small populations that better survey rugged fitness landscapes and escape suboptimal fitness peaks[42,43]. The strain, the host, and prior immunity will all influence which processes are at play during influenza virus infection.

Immune pressure positively selects antigenically advanced influenza virus variants on the global scale[1]. Yet, these same variants rarely emerge in an acutely infected host, where infections are dominated by purifying selection and low intra-host diversity[4,5,7,8,44]. The apparent disconnect between influenza virus evolution on the global versus individual scale is incompletely understood. One potential contribution comes from long-term or chronic infections in immunocompromised individuals. Antigenic variants, with some mutations matching those found in future drifted strains, emerged in a cohort of four immunocompromised cancer patients who had extraordinarily prolonged H3N2 influenza virus infections[45]. Our findings reveal that intrahost bottlenecks are major contributors to the limited evolution detected in an infected individual. This may partially explain why highly pathogenic avian H5N1 strains have not yet acquired the ability to transmit within people. H5N1 strains infect the lower respiratory tract in humans, where the preferred α2,3-sialic acid receptors are more abundant[14,15]. We saw little evidence of mixing between compartments in the lower respiratory tract. Thus, while only a few mutations are needed to confer airborne transmission to H5N1 strains in experimental settings, the spatial structure and compartmentalization of infection in humans may impose bottlenecks that prevent fixation of variants and migration to the upper respiratory tract, let alone transmission to a new host[10,11].

Intrahost bottlenecks are common features during viral infections. Physical barriers establish bottlenecks as enteroviruses escape the gut in mouse models[46,47]. Poliovirus then faces another bottleneck related to IFN responses that restrict diversity as the virus invades the central nervous system[46,48,49]. Our data do not reveal whether the bottlenecks we describe for influenza virus are due to physical barriers, innate immune response, or other factors. Similar forces shape evolution in arthropod vectors. Repeated bottlenecks winnow diversity as the virus moves between different tissues within mosquitoes for West Nile virus, Venezuelan equine encephalitis virus, and Zika virus[41,50,51]. Error-prone replication then repopulates diversity in the new sites of replication. Repeated bottlenecks, such as those we have identified during intra-host dissemination, decrease diversity and may reduce fitness. Yet, they also act to purge deleterious mutations and facilitate escape from local fitness maxima and exploration of additional evolutionary space. The opposing effects of these bottlenecks shape the kinetics of intra-host influenza virus evolution.

In summary, we use a very large viral population with uniquely addressable members to demonstrate the existence of multiple bottlenecks during the dissemination of the influenza virus throughout the respiratory tract. We posit that these bottlenecks contribute to the limited impact of positive selection on intrahost evolution. Moreover, they provide additional barriers to initial cross-species transmissions and sustained transmission once a virus spills over into a new host. Coupled with the stringent bottlenecks that occur during inter-host transmission, our results help explain the stochastic nature of influenza virus evolution at the local scale.

## Methods

**Cells**. MDCK cells (ATCC), MDCK-SIAT1-TMPRSS2 cells[52], and 293T cells (ATCC) were maintained in Dulbecco's modified Eagle medium (DMEM) supplemented with 10% heat-inactivated fetal bovine serum (Atlanta Biologicals), 100 μg/mL streptomycin, and 100 U/mL penicillin at 37 °C and 5% CO$_2$.

**Generation of pHW2000-all-ΔPA-ΔHA A/California/07/2009**. Bidirectional cassettes for CA07 PB2, PB1, NP, NA, M, and NS gene segments were sequentially amplified and inserted into a pHW2000 vector via Gibson assembly. The resulting 17.3 kb plasmid was sequence verified. Annotated plasmids sequences can be found at https://github.com/mehlelab/barcoded_flu_analysis.

**Generation of barcoded PA**. Barcodes were inserted into the previously described PASTN rescue construct that expresses a polyprotein encoding PA and Nanoluciferase separated by the 2 A cleavage site[22,53]. The barcode was originally synthesized (IDT) as a single-stranded DNA oligo containing ten random nucleotides flanked by sequence to enable cloning immediately downstream of the Nanoluciferase open reading frame (Supplementary Table 1). The oligo was amplified in 25 individual low-cycle PCRs using Q5 polymerase (NEB). The amplicons were pooled and gel purified prior to cloning into a modified PASTN plasmid using 50 individual ligation reactions. Ligations were transformed into Mach1 competent *E. coli* (Thermo Fisher), plated on LB agar supplemented with 200 μg/mL ampicillin, and grown overnight at 37 °C to yield 220,000 transformants, indicative of theoretical maximum library size. Colonies were scraped off plates, collected in 750 mL liquid LB with ampicillin, and grown for 3 hr at 37 °C. DNA was then purified using the Zymo Midiprep kit to create the plasmid library. Plasmid stocks were deep sequenced to determine library size and diversity.

**Generation of barcoded HA**. *HA* from A/California/07/2009 was cloned into the pHW2000 rescue plasmid. Silent mutations were introduced into the final 25 amino acids of the open reading frame and 80 nt from the 3′-end of the open reading frame were repeated downstream to recreate a contiguous packaging signal[23]. Initial libraries utilized the native HA sequence, whereas subsequent libraries included the tissue-culture adaptive K153E (nt A540G in cRNA) mutation[32]. A single-stranded oligo (IDT) was synthesized containing a randomized 10-nt barcode as well as a 6-nt registration mark, either GCTAGC (NheI) or CTGCAG (PstI) (Supplementary Table 1). This registration mark is a conserved and identifiable region for all the individuals within a given library. Barcodes were amplified in 50 low-cycle PCRs and cloned following the same strategy as for PA libraries. Approximately 60,000 transformants were obtained with the NheI registration mark, and 60,000 for the PstI registration mark. Plasmid stocks were deep sequenced to determine library size and diversity.

**Rescue of dual-barcoded virus libraries**. Barcoded CA07 virus libraries were rescued via reverse genetics using the pHW2000-all-ΔPA-ΔHA and PA and HA barcoded plasmids described above. Briefly, 293T cells were forward transfected with 2.7 μg pHW2000-all-ΔPA-ΔHA, 450 ng pHW2000-PASTN-barcode library, 450 ng pHW2000-HA-barcode library (NheI or PstI variant), and 400 ng pHAGE2−EF1αInt−TMPRSS2−IRES−mCherry-W[52] in a 6-well format. Plasmids were combined with 200 μL jetPRIME Buffer and 8 μL jetPRIME reagent (Polyplus) per well. 120 independent transfections were performed per viral library. 24 hr post-transfection, media was removed and cultures were overlaid with MDCK-SIAT1-TMPRSS2 cells in OptiVGM (OptiMEM supplemented with 0.3% bovine serum albumin, 100 μg/mL calcium chloride, 100 μg/mL streptomycin, and 100 U/mL penicillin). Rescue viruses were harvested 48–72 hr later and pooled based on their registration mark. Viruses were amplified on 20, 15 cm dishes of MDCK-SIAT1-TMPRSS2 cells infected at an MOI ~0.01 in OptiVGM for 66 hr. A total of at least 10$^6$ infectious virions spread across the 20 plates were used to amplify each stock. Viruses were pooled based on their registration mark and cellular debris was removed by centrifugation. Viral titers were determined by plaque assay and TCID50Glo assays on MDCK and MDCK-SIAT-TMPRSS2 cells[54].

**Library preparation for barcoded amplicons**. Viral RNA was extracted from all samples using the Maxwell RSC Viral Total Nucleic Acid Purification Kit (Promega) according to the manufacturer's instructions. RNA was subjected to DNAse treatment using the TURBO DNAse (Invitrogen), and reverse transcribed in 20 μl using the SuperScript IV VILO master mix (Invitrogen) with PA and HA gene segment-specific primers (Supplementary Table 1). DNA amplicons for HA and PA gene segments with partial sequencing adapters were generated via PCR amplification of cDNA using the Phusion High-Fidelity DNA Polymerase (New England BioLabs) and gene segment-specific primers (Supplementary Table 1). To minimize technical bottlenecks during library preparation, reverse transcription

and the first PCR amplification for all samples were performed in triplicate and pooled together prior to DNA purification. PCR products from mouse samples were gel purified whereas PCR products from ferret samples were purified with paramagnetic beads using the AMPure XP for PCR Purification kit (Beckman Coulter). Purified PCR products were used in a second PCR reaction for incorporating sample-specific 5′-end indexes and additional Illumina sequencing adapters (Supplementary Table 2). Final PCR products were gel purified and individual DNA concentrations were determined with the Qubit dsDNA High Sensitivity Assay Kit on the Qubit Fluorometer (Invitrogen). Samples were quality controlled using the Bioanalyzer High Sensitivity DNA Analysis Kit and the Agilent 2100 Bioanalyzer (Agilent). All samples were prepared and sequenced in technical replicate. Detailed protocols and code were available at https://github.com/mehlelab/barcoded_flu_analysis.

**Library preparation for whole-genome sequencing**. Library preparation was similar to our prior approaches[55]. Briefly, viral RNA was extracted from all samples using the Maxwell RSC Viral Total Nucleic Acid Purification Kit (Promega) according to the manufacturer's instructions. RNA was subjected to DNAse treatment using the TURBO DNAse (Invitrogen), and reverse transcribed in 20 μl using the SuperScript IV VILO master mix (Invitrogen) with the Uni12 primer (Supplementary Table 1) that targets conserved ends of all gene segments. Segments were amplified by PCR with gene-specific primers (Supplementary Table 1), gel purified, and DNA concentrations were determined using the Qubit dsDNA High Sensitivity Assay Kit on the Qubit Fluorometer (Invitrogen). 1 ng of each segment was pooled and used as input for the Nextera DNA Library Prep kit where samples were tagmented and indexed according to the manufacturer's instructions. Tagmented and amplified products were purified with AMPure XP paramagnetic beads for PCR Purification kit (Beckman Coulter) in two consecutive steps (0.5x and 0.7x) and were quantified using Qubit dsDNA high-sensitivity kit (Invitrogen, USA). Sample quality control was performed using the Bioanalyzer High Sensitivity DNA Analysis Kit in the Agilent 2100 Bioanalyzer (Agilent). All samples were prepared and sequenced in technical replicate. Detailed protocols can be found at https://github.com/mehlelab/barcoded_flu_analysis.

**Deep sequencing**. Amplicon and whole-genome libraries were sequenced on the Illumina MiSeq system using the MiSeq Reagent Kit v2-500 and v3-600, respectively (Illumina). Amplicon and whole-genome samples that passed quality control were pooled in a 4 nM library with nuclease-free water. 5 μl of the 4 nM library pool was denatured with 5 μl 0.2 N of NaOH and diluted using the HT1 Hybridization Buffer (Illumina) to a concentration of 8 pM for amplicon samples and 10 pm for whole-genome samples. A PhiX library was prepared similarly and added at 30% of the input for amplicon sequencing and 1% for whole-genome libraries. Samples were loaded on the respective MiSeq cartridge and paired-end sequencing reads were generated (Illumina).

**Sequencing data analysis for barcoded amplicons**. We generated a custom bioinformatics pipeline to process raw FASTQ files and quantify barcode frequencies (https://github.com/mehlelab/barcoded_flu_analysis). Briefly, raw FASTQ paired reads were demultiplexed, merged, and aligned to a custom amplicon reference containing barcode and registration mark regions as strings of N's (BBMap Tools v38.87). Reads with lengths the same as the average insert size was aligned, sorted, and indexed with Samtools (v1.11, htslib v1.11). BAM files with aligned reads were processed and trimmed to the region containing the registration mark and/or the barcode sequence using command-line tools (Seqtk v1.3, Bash v3.2.57).

Sequencing of the invariant registration mark was used to benchmark the fidelity of our sequencing. Over 99.1% of reads were perfect matches, where most differences were the result of a single nucleotide change from the expected sequence. Therefore, to correct for amplification and/or sequencing errors that may inflate the number of unique barcodes, we used UMI-tools (v1.1.1) to generate consensus barcode clusters with reading counts via the adjacency network-based clusterer method[27]. Parental barcodes and their apparent mutational offspring were clustered prior to enumeration and cluster frequencies were generated and visualized via a custom Python pipeline (Python v3.8.5, Pandas v1.1.3, Matplotlib v3.3.2, Numpy v1.19.2) or in Prism 9. Manual inspection of sequences in the raw FASTQ files confirmed these as bona fide barcodes and not the product of misalignment.

Rarefaction-extrapolation was performed with iNEXT (v2.0.20)[56,57]. The total number of unique barcodes and total reads across all samples was used to establish average read depth for NheI- or PstI-barcoded stocks. These were used to parameterize Poisson sampling in R (v4.0.2) to simulate data from an ideal stock where barcodes are equally distributed.

**Sequencing data analysis for whole gene segments**. We generated a custom bioinformatic pipeline to process raw FASTQ files and determine SNV from our barcoded viral samples (https://github.com/mehlelab/barcoded_flu_analysis). Briefly, raw FASTQ paired reads were demultiplexed and downsampled to 2000 reads (BBMap Tools v38.87). We mapped reads to the full IAV genome using a Burrows-Wheeler alignment (BWA v.0.7.17), used Samtools (v1.11, htslib v1.11) to

sort aligned reads, and called variants using LoFreq (v2.1.5) with a minimum coverage of 500 reads, the base call quality of at least 30 and a frequency exceeding 0.03 (3%). Our reference IAV genome included barcode insertions in PA and HA, registration marks on HA as a string of 10 N's, NanoLuc inserted in PA, and repeated packaging signals. SNVs were annotated using SnpEff (v5.0) to determine the impact of each variant on the amino acid sequence, and the resulting variant call format files were manipulated using bcftools (v1.11) to transform into user-defined formats. Plots were generated using custom bioinformatic pipelines in Python language (Python v3.8.5, Pandas v1.1.3, Matplotlib v3.3.2, Seabornv0.11.2, and Numpy v1.19.2).

**Long-read sequencing of HA and analysis**. Viral RNA was made compatible for sequencing on an Oxford Nanopore Technologies instrument using the 1D PCR Barcoding Kit. Briefly, viral RNA was extracted using the Maxwell RSC Viral Total Nucleic Acid Purification Kit (Promega), subjected to DNAse treatment using TURBO DNAse (Invitrogen), and reverse transcribed and amplified using the SuperScript IV One-Step RT-PCR system with HA-specific primers. Amplified DNA was gel purified using the QIAquick Gel Extraction Kit (Qiagen) and quantified using the Qubit dsDNA High Sensitivity Assay Kit on the Qubit Fluorometer (Invitrogen). Normalized samples were made compatible for long-read sequencing using the 1D Native Barcoding ONT Kit (SQK-LSK-109), following the manufacturer's protocol. Libraries were loaded onto a flow cell and run on the ONT GridION machine. Bases were called in real-time using the ONT software package Guppy 3.2.6. Minimap2 was used to map reads to the influenza virus HA segment and discard low-quality reads[58]. The Sam2Tsv (v1.0) module from Jvarkit[59] was used to convert the sam to a tsv file so Pandas could be used to extract the bases at position 540 (NheI library) or 543 (PstI library) and the barcode sequence. To account for sequencing error associated with ONT that may inflate the number of unique barcodes, UMI-tools (v1.1.1) was used to generate consensus barcode clusters prior to analysis. Given the higher error rate associated with ONT, the more conservative directional network-based clusterer method was used[27] Results were visualized with Prism 9.

**Growth kinetics**. Triplicate dishes of confluent MDCK-SIAT1-TMPRSS2 cells were inoculated at an MOI of 0.01 with virus diluted in OptiVGM. Virus was adsorbed for 1 hr at 37 °C, removed from cells, and replaced with fresh OptiVGM. Virus was sampled over time and titered by plaque assay on MDCK-SIAT1-TMPRSS2 cells.

**Mouse infections**. All mouse experiments were approved by the University of Wisconsin Madison Institutional Animal Care and Use Committee. Mice were housed at 22 °C and ~30% relative humidity with a 12 hr light-dark cycle. In all, 18 9-week-old female BALB/c mice (Charles River Labs) were randomly divided into groups of 6 to receive either WT HA, HA-K153E-NheI-bc, or HA-K153E-PstI-bc virus. All viruses contained barcoded PASTN. Mice were inoculated intra-nasally with $10^5$ TCIDGlo50 of virus in 35 μL media. Mice were weighed daily and monitored for clinical signs of infection. Three animals from each group were sacrificed at 3 dpi and the remainder at 6 dpi. Lungs were removed, dounce homogenized in 1× DPBS and clarified at $2000 \times g$ for 5 min. The clarified homogenate was titered via plaque assay and TCIDGlo50 assay on MDCK-SIAT1-TMPRSS2 cells. Viral RNA was recovered from homogenate and sequenced as described above.

**Ferret infections**. All ferret experiments were approved by the St Jude Children's Research Hospital Animal Care and Use Committee. 12-week old male ferrets (Triple F Farms, Sayre, PA) confirmed to be seronegative for influenza virus were housed individually. Each animal was infected intranasally following previous approaches that ensure site-specific inoculation of the upper respiratory tract[37]. Animals were inoculated with $5 \times 10^5$ PFU HA-K153E-NheI-bc with barcoded PASTN diluted in PBS (Corning) containing 100 U/mL penicillin and 100 μg/mL streptomycin (Corning) in a total volume of 500 μl. Nasal washes were collected at 1, 3, and 5 dpi. Briefly, animals were anesthetized with 0.25 mL ketamine and nasal washes were collected by administering 1 mL PBS/pen-strep dropwise onto the nostrils. Animals were sacrificed at 5 dpi and trachea and separated lung lobes were removed and frozen prior to processing. The tissue was homogenized in 10% (w/v) L-15 media using an OMNI TH220-PCRD homogenizer and clarified at $500 \times g$ for 10 min to remove cell debris. Viral titers in homogenates and nasal washes were determined by TCIDGlo50 assay. Viral RNA was recovered from homogenate and sequenced as described above.

**Statistical analysis**. Viral titers are presented as the mean of $n = 3$ and significance was tested with a two-way ANOVA with Tukey's post hoc analysis. Replicate sequencing runs were analyzed with Pearson's and Spearman's correlation coefficients (Prism 9). Population diversity and evenness were assessed by

measuring Shannon's diversity (H')[30] as follows:

$$H' = -\sum_{i=1}^{S} p_i \ln p_i \tag{1}$$

The Gini-Simpson diversity index ($H_{GS}$) was calculated as:

$$H_{GS} = 1 - \sum_{i=1}^{S} p_i^2 \tag{2}$$

For both diversity calculations, $S$ is the total number of barcodes detected (richness) and $p_i$ is the frequency of the $i$-th barcode in that sample.

Population evenness is bounded at 0 and 1 and defined as the actual barcode diversity divided by the maximum possible diversity ($H'_{max} = \ln(\text{richness})$) for the sample[29]:

$$\text{evenness} = \frac{H'}{H_{max}} \tag{3}$$

Bray–Curtis dissimilarity ($BC_{ij}$) was used to assess compositions and compare lung lobes[60], defined as:

$$BC_{ij} = 1 - \frac{2C_{ij}}{S_i + S_j} \tag{4}$$

where $C_{ij}$ is the sum of the lesser values for only barcodes found in both lobes. $S_i$ and $S_j$ are the total number of barcode reads detected in either lobe.

**Multinomial model of bottleneck size.** While several methods currently exist that estimate viral transmission bottleneck sizes between a donor sample and a recipient sample, none of these methods are appropriate for estimating $N_b$ from the viral lineage frequencies available from this study. The beta-binomial approach outlined in Sobel-Leonard et al.[61] assumes that each locus is biallelic and unlinked to other loci. A more recent approach developed by Ghafari et al.[62] allows for linked loci by reconstructing haplotypes. However, the number of haplotypes that can be considered in their approach is very low relative to the number of viral lineages observed in this study. We thus developed a simple statistical approach to estimate $N_b$ that allows for a large number of observed haplotypes (here, viral lineages). The approach is as follows:

1. Identify the set of lineages observed in the donor sample and calculate their frequencies
2. Calculate the overall number of unique viral lineages present in the recipient sample that is also found in the donor sample
3. Over a range of $N_b$ values, for each $N_b$ do as follows:

   a. Draw $n$ separate times from a multinomial distribution with the probability vector given by the viral lineage frequencies in the donor sample and the number of trials being the proposed bottleneck size $N_b$. A given draw can be considered a random realization of viral lineages in a recipient sample seeded by a bottleneck size of $N_b$. We let $n = 500$; higher values of $n$ did not appreciably alter the results.
   b. For each of the $n$ draws from the multinomial distribution, quantify the number of viral lineages present in the simulated recipient sample.
   c. Calculate the mean and standard deviation of the number of viral lineages present in the $n$ simulated recipient samples.
   d. Calculate the probability of observing the observed number of viral lineages in the recipient sample using a normal distribution with mean and standard deviation as calculated in step 3c. The log of this probability yields the log-likelihood of the bottleneck size being $N_b$.

4. Identify the maximum likelihood estimate of $N_b$ as the $N_b$ yielding the highest log-likelihood. Calculate the 95% confidence interval of $N_b$ as the set of $N_b$ values that yield log-likelihood values within 1.92 log-likelihood units of the maximum likelihood estimate of $N_b$.

Note that the bottleneck size $N_b$ that is estimated through this procedure quantifies the number of viral particles that seed the recipient tissue. It does not quantify the number of viral particles that establish genetic lineages in the recipient tissue.

Forward simulations of the multinomial model were performed by setting $N_b$ to its maximum likelihood value, performing a single multinomial draw as in step 3a, and plotting the resulting simulated frequencies of the viral lineages in the recipient sample against those observed in the donor sample. Forward simulations of the multinomial model with environmental noise were performed by first similarly setting $N_b$ to its maximum likelihood value and performing a single multinomial draw. Unlike in step 3a, however, the probability vector was instead given by perturbed viral lineage frequencies from the donor sample. Perturbation of lineage frequencies was done by multiplying each lineage frequency by exp($E$), and then normalizing, where $E$ is normally distributed random variable with mean 0 and standard deviation σ. Perturbing lineage frequencies in this manner implicitly capture stochastic replication dynamics within the recipient tissue without the need to simulate viral growth explicitly. As in the previous forward simulations, the resulting simulated frequencies of the viral lineages in the recipient sample were then plotted against the observed viral lineage frequencies from the donor sample.

We considered a range of σ values. With σ = 0, the original multinomial model described above is recovered, leading to strong correlations between viral lineage frequencies in the nasal wash and the lung lobes that are not empirically observed. At higher values of σ, model-projected evenness and diversity decreased, and the projected correlations between lineage frequencies in the nasal wash and the lung lobes became weaker. Here, we let σ = 2 for each of the lung lobes, based on simulation results that indicated that this level of replication heterogeneity could recover the weak (but apparent) observed correlation between nasal wash and lung lobe lineage frequencies.

**Reporting summary**. Further information on research design is available in the Nature Research Reporting Summary linked to this article.

## Data availability
All sequencing files have been deposited as BioProjects PRJNA746307, PRJNA746319, and PRJNA746317 with details and SRA accessions in Supplementary Table 3. Source data are provided in this paper.

## Code availability
Code can be found at https://github.com/mehlelab/barcoded_flu_analysis

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

## Acknowledgements

We thank Jesse Bloom and Yoshihiro Kawaoka for providing key plasmids and reagents. We thank Christopher Brooke and Brigette Martin for their assistance in virus rescue protocols. This work is funded by AI125392 to A.M. and T.C.F. and by the National Institute of Allergy and Infectious Diseases under HHS contract HHSN27220140006C for the St. Jude Center of Excellence for Influenza Research and Surveillance, NIH grant R01AI140766, and ALSAC to S.S.C. K.A.A. is supported by GM007215. K.M.B. is supported by AI145182. L.A.H. is supported by HG002760. G.A.S. is supported by a Rath Foundation Wisconsin Distinguished Graduate Fellowship and AI007414. A.M. holds an Investigators in the Pathogenesis of Infectious Disease Award from the Burroughs Wellcome Fund and is an H. I. Romnes Faculty Fellow funded by the Wisconsin Alumni Research Foundation.

## Author contributions

K.A.A., L.A.H., K.M.B., K.K., S.S.C., T.C.F., and A.M. contributed to the conceptualization. K.A.A., L.A.H., K.M.B., K.K., T.C.F., and A.M. developed methodology. K.A.A., L.A.H., K.M.B., K.K., and A.M. wrote or executed software. K.A.A., L.A.H., K.M.B., V.M., BL, RH, G.A.S., K.K., and A.M. performed validation. K.A.A., L.A.H., K.M.B., K.K., T.C.F., and A.M. performed formal analysis. K.A.A., L.A.H., K.M.B., V.M., B.L., R.H., G.A.S., E.B., C.A.H., G.L.B., K.K., and A.M. did the Investigation. K.A.A., L.A.H., and K.M.B. were responsible for data curation. K.A.A., L.A.H., K.K., T.C.F., and A.M. wrote the original draft, with all authors reviewing and editing. K.A.A., L.A.H., K.M.B., K.K., and A.M. created Visualizations. S.S.C., T.C.F., and A.M. supervised the work with T.C.F. and A.M. as project administrators. K.A.A., L.A.H., K.M.B., G.A.S., S.S.C., T.C.F., and A.M. acquired funding.

## Competing interests

The authors declare no competing interests.
