## [Peer Review File · Nature Communications]

Reviewer comments, first round of review

Reviewer #1 (Remarks to the Author):

This is a cool paper that looks at intra-host population diversity and bottlenecks for influenza virus. Although there have been quite a few studies of such processes at the level of transmission, there have been fewer on within-host bottlenecks. Overall, I found this study very interesting, and strongly support its publication although I do have a number of relatively minor suggestions.

SPECIFIC COMMENTS

- The Introduction is good, but it might be worth briefly noting that the described drift-dominated evolution applies to normal short acute infections, but that there is evidence of clear within-host selection in much rarer chronic infections. PMID 28653624
- Line 87-89: "individuals" should be "individual virions". Also, might be good to re-explain why this is true: many virions will not have genetically unique markers.
- I think it's important to mention the Varble et al paper (PMID 25456074) as prior work that has also used barcodes to look at transmission bottlenecks. Right now it's cited, but not clearly called out as a prior use of a similar approach.
- The arguments on lines 96-99 about why HA barcodes might reflect selection whereas PA should represent population size rely on the assumption that there are extremely high levels of within-host reassortment that breaks up linkage disequilibrium between mutations in HA and those in other segments. This assumption is uncertain, since there is substantial evidence of within-host hitchhiking and relatively low rates of reassortment.
- Line 100: is there a difference between "packaging" and "bundling" signals? I've never actually heard people refer to the latter.
- Lines 117-118: In addition to just counting unique barcodes, it would be more informative to show a rarefaction curve, which would indicate how balanced the barcodes are. In other words, when you sample X sequencing reads, what is the number Y of unique barcodes observed. This can then be compared to the theoretical distribution under Poisson sampling of barcodes with equal weights.
- Paragraphs from line 117 to 142, the authors should clearly state at what MOI the passaging was done and the total number of infectious particles passaged. This is a key detail so needs to be described in Results.
- For the section on selection of the HA K153 or K154 mutations in the original bottlenecked libraries, could the authors comment more on the magnitude of the sweep? I'm not surprised that cell-culture adaptations like this spread, but the mutations appear to be going from frequencies of ~0.001% to ~50% in a single passage, which is ~50,000-fold amplification. Usually these types of cell-culture adaptations increase viral titers ~10-fold, so I'm surprised the extent of enrichment is so dramatic over just a single passage. Could this be commented on or explored more? In particular, do they have direct measurements of how much K153E improves growth?
- Supp Figure 1 legend: says "all for replicates" when it should be "four".
- Lines 162-165: this is **extremely confusing**, and remains so even after looking at Supp Figure 1. I think what the authors are saying is that most sequencing reads correspond to shared barcodes, but that not all detected barcodes are shared. This is expected if some barcodes are really very low frequency or are actually just sequencing errors. This point does also call into some question the relevance of the "unique barcodes" metric in Fig 3A if many of the barcodes are actually really low frequency or are sequencing errors. Accordingly, Fig 3 should provide

comparable data to Fig 2 on the evenness of the barcodes before and after passaging.

- It's not clearly explained if Figure 3D is showing pre- or post-passaging data for the new libraries. Also, what is the deal with the mutation near the C-terminus of HA? Is this another putative adaptive mutation?

- Figure 4C, I don't like the "total" barcode metric. Due to the various issues about barcode skewing, sequencing errors, etc I expect this metric will be susceptible to things like different sequencing depths and isn't a good indicator. How about using something like the inverse Simpson index, which calculates the effective diversity in a way that isn't susceptible to skewing, sequencing errors, depth, etc. In fact, I think the inverse Simpson index would be a more intuitive measure than the Shannon diversity in all cases as it can be directly interpreted as a measure of effective number of unique barcodes. The Shannon diversity doesn't have a natural numerical interpretation and depends on the log base you choose.

- Line 217: related to a point I made above, I don't think that's very surprising. I think there is quite a bit of evidence that the effective rate of intra-host reassortment isn't all that high.

- I actually found Supp Figures 4 and 5 quite a bit more intuitive than much of what is plotted in Figure 6 and especially Figure 5 for showing the differences in variants among anatomical sites. The current presentation is adequate, but do consider adding some simpler more direct plots to main figures. The difference between anatomical sites is the coolest part of the paper, so it's worth making it as clear

Reviewer #2 (Remarks to the Author):

Amato and colleagues added short barcodes to two influenza A virus genome segments, and went to great lengths to reconstitute virus populations with high levels of neutral barcode variation. These populations were then inoculated into the upper respiratory tract of both mice and ferrets, and population diversity was monitored as the virus was transmitted to the trachea and lungs. In mice, there was not clear evidence of population bottlenecks, whereas in ferrets there was clear evidence for population bottlenecks as the virus invaded the trachea and different lung lobes. The authors conclude that narrow population bottlenecks are an important feature of the within-host infection dynamics in the clinically relevant host (ferrets), and that these strong random effects may help to explain why there appears to be limited adaptive evolution within hosts.

This is an excellent and timely study on the within-host bottlenecks in virus populations, and an experimental tour de force. The authors have gone to great lengths to set up the system carefully, and also to provide a balanced and transparent presentation of the data and results. Below we have some questions about certain aspects of the study and comments on the presentation, which we hope the authors will find useful for improving the manuscript.

Main comments:

The way the barcoding approach was implemented is excellent from a technical perspective. Repeating the barcoding for the preadapted variant was a simple but effective solution. One potential downside to this approach is the "background" signal. The authors have clearly taken some steps to mitigate the impact of sequencing errors (i.e., lines 473-481 in the methods section), but there is no validation of this approach and it does not guarantee all barcode variants are bona fide. We don't think this comes as any surprise to anyone who has worked with high-throughput sequencing (HTS) and we think the presentation of the data is fair, but we don't feel the authors have fully considered the implications of sequencing error in the interpretation of the results. For example, the data suggest that some variants detected are low-frequency false positives (I.e., Fig. S6, where the "lung only" variants must be (i) sequencing errors, (ii) de novo mutations, or (iii) variants missed in the wash). The observed patterns in lung lobes, one or a few variants at high frequency (e.g., Fig. 5I MR and LR, particularly for PA), suggest that one barcode was fixed here and the remaining signal is largely background. Do the authors agree? Is it possible

to distinguish between the explanation suggested by the authors for these patterns (lines 294-296) and sequencing errors that remain unidentified? Because these issues are inherent to HTS and HTS is by far the best approach for characterizing viral populations, this issue is unavoidable unless CirSeq is used. But we do feel it can be better taken into account in the interpretation and presentation of the results.

The proposed explanation (lines 294-296) for the main mechanisms that underlie the observed distribution of genotypes is intuitive and we fully agree it is highly relevant. Although this explanation is pivotal for understanding these patterns, it is a pity the presentation doesn't come full circle by showing quantitatively that this explanation works. Ideally, one would estimate the dispersion in "takeoff" timepoints that best accounts for the data, but the authors would probably counter that this is beyond the scope of this presentation (and we would tend to agree). As a simple solution and presentation tool, therefore we suggest that the authors add a "seeding time" component to the simulations presented in Figure S5, to show that this mechanism can - in principle - lead to patterns that are more congruent with the data. We also think it is also important to provide some details about the model described here: the approach appears to be very straightforward, but it would be good to describe it explicitly.

Minor comments:

Line 27: "Bottlenecks stochastically sampled.." should be "Bottleneck events stochastically sampled.."

Lines 29-31, 73-78 and elsewhere: Random events (i.e., small mutation supplies and population bottlenecks) can also be advantageous to evolution, when sign epistasis is common and populations can become trapped on suboptimal peaks (E.g., PNAS 112(24): 7530-7535, PNAS 114(48): 12773-12778). In fact, one could argue that the infection dynamics described here could be quite beneficial for long-term evolvability and effectively reconnoitering a rugged fitness landscape. Therefore, for a complete and balanced interpretation of the results, I would also consider this perspective on viral evolution. We realize the authors may not agree, but then we would argue why they feel this scenario is less relevant.

Lines 33-41: Paradoxical. So at what level does selection then act? Primarily on the between-hosts level? They need to explain this further for a broad audience.

Lines 43-57: At the authors' discretion: having an entire paragraph on H5N1 adaptation to mammals suggests that this work would also involve this strain or the adaptive process. It might be helpful to point out right away this is just an example to highlight the importance of evolutionary constraints to adaptation?

Line 48: The mutations that impede evolution/block trajectories need not be "deleterious" as such. They also can be beneficial or neutral, what matters is epistatic interactions (i.e., to be incompatible with other beneficial mutations).

Lines 60-62: High cellular MOI can also constrain evolution if the benefits (or costs) of mutations act in trans (E.g., J Virol 84(4):1828-37), by disassociating the cost/benefit from the genome precluding within-cell selection. Although there are evolutionary advantages to high MOIs, clearly there are also potential disadvantages. Useful to mention this here to make the argument more balanced.

Lines 96-97: Mention explicitly that the "result of selection" would be hitchhiking of barcoded variants with beneficial mutations?

Line 136: What was the effective population size here? Is it possible to provide an expectation a priori?

Figures 2/3: Very informative figures, but it would be cool to display the diversity in the labelled variants made after preadaptation as is done in Fig. 2b. The contrast would be striking and it would make the results very intuitive (whilst we realize that Fig. 3 is technically more informative). Panels 5 f/g: It took a while to figure out that these are the tracheal samples. Make this clearer in the legend, and/or even better, add a visual cue in the figure itself?

Conceptually, since the authors anticipated tight bottlenecks, one could question whether this barcoding approach was needed. I.e., Could experimental results with a few markers (I.e., mixtures of NheI and PstI) have rendered the same result (i.e., maintenance of both variants with minimal changes in frequency nasally, followed by the consistent fixation of one variant in the trachea or lungs [for ferrets])? We would find this an interesting discussion point, and others contemplating similar work could greatly benefit from this.

Line 454: Following the link leads to a "Link not found, 404 Error" message. Should this be the same link as line 438?

Supplemental Figure 1B: change "for" to "four".

REVIEWER COMMENTS

Reviewer #1 (Remarks to the Author):

This is a cool paper that looks at intra-host population diversity and bottlenecks for influenza virus. Although there have been quite a few studies of such processes at the level of transmission, there have been fewer on within-host bottlenecks. Overall, I found this study very interesting, and strongly support its publication although I do have a number of relatively minor suggestions.

We thank the reviewer for their supportive and the constructive criticisms below that have strengthened the resubmission.

SPECIFIC COMMENTS

- The Introduction is good, but it might be worth briefly noting that the described drift-dominated evolution applies to normal short acute infections, but that there is evidence of clear within-host selection in much rarer chronic infections. PMID 28653624

We agree. The contribution of long-term infection in immunocompromised individuals is intriguing, and this is now included in the discussion as a potential way to bridge the gap between global and local evolution (lines 354-358).

- Line 87-89: "individuals" should be "individual virions". Also, might be good to re-explain why this is true: many virions will not have genetically unique markers.

Suggested changes are included on line 90.

- I think it's important to mention the Varble et al paper (PMID 25456074) as prior work that has also used barcodes to look at transmission bottlenecks. Right now it's cited, but not clearly called out as a prior use of a similar approach.

Line 103-104 explicitly acknowledged prior work by Varble, et al. that used barcoding to monitor bottlenecks.

- The arguments on lines 96-99 about why HA barcodes might reflect selection whereas PA should represent population size rely on the assumption that there are extremely high levels of within-host reassortment that breaks up linkage disequilibrium between mutations in HA and those in other segments. This assumption is uncertain, since there is substantial evidence of within-host hitchhiking and relatively low rates of reassortment.

We agree that the premise relies on high levels of reassortment. That was our rationale, yet as detailed later in the text (and alluded to by the comment above), this is not what we detected.

- Line 100: is there a difference between "packaging" and "bundling" signals? I've never actually heard people refer to the latter.

Packaging sequences are those within a vRNP required for efficient incorporation into virions, whereas bundling sequences are proposed by Goto et al. 2013 (PMID 23926345) to mediate

inter-vRNP interactions to ensure virions contain a full complement of viral genes. We now included the Goto 2013 citation.

- Lines 117-118: In addition to just counting unique barcodes, it would be more informative to show a rarefaction curve, which would indicate how balanced the barcodes are. In other words, when you sample X sequencing reads, what is the number Y of unique barcodes observed. This can then be compared to the theoretical distribution under Poisson sampling of barcodes with equal weights.

Thank you for this suggestion. We analyzed barcode distributions in our pre-adapted virus libraries with rarefaction-extrapolation curves and report these data in Sup Fig 1E and 2B and lines 177-182 with methods on line 501-504. We also used Poisson sampling to simulate data from a perfectly even population. These data were used to generate rarefaction-extrapolation for an ideal population. Consistent with our earlier analyses, the new results reveal that we can accurately and repeatedly measure population diversity, that our populations are diverse, yet they differ from ideal populations and are not perfectly distributed.

- Paragraphs from line 117 to 142, the authors should clearly state at what MOI the passaging was done and the total number of infectious particles passaged. This is a key detail so needs to be described in Results.

Viruses were amplified on 20 15-cm dishes of MDCK-SIAT1-TMPRSS2 cells infected at an MOI ~0.01 in OptiVGM for 66 hr. A total of at least 10^6 infectious virions spread across the 20 plates were used to amplify each stock. These conditions were selected to minimize the impact of technical bottlenecks. Details are included on lines 139 and 433-435.

- For the section on selection of the HA K153 or K154 mutations in the original bottlenecked libraries, could the authors comment more on the magnitude of the sweep? I'm not surprised that cell-culture adaptations like this spread, but the mutations appear to be going from frequencies of ~0.001% to ~50% in a single passage, which is ~50,000-fold amplification. Usually these types of cell-culture adaptations increase viral titers ~10-fold, so I'm surprised the extent of enrichment is so dramatic over just a single passage. Could this be commented on or explored more? In particular, do they have direct measurements of how much K153E improves growth?

We too were surprised by the selective advantage afforded by the HA mutations. Our viral amplification proceeded for 66hr. Assuming the replication cycle in MDCK-SIAT-TMPRSS2 cells is ~8hr, this would be over 8 infectious cycles. MDCK-SIAT-TMPRSS2 cells are highly susceptible to influenza virus, and the replication cycle may be even shorter. The burst size for influenza virus in MDCK cells is estimated between 1,000-10,000 particles (Stray and Air 2001; 11451482). Thus, even small increases in fitness may have outsized effects. Prior work by Chen, et al. 2010 (19864389) showed that viruses encoding HA K153E or HA K154E grow faster to produce a larger plaque phenotype and yielded almost 10-fold more virus compared to WT when grown in eggs. This combination may account for the pronounced sweep.

- Supp Figure 1 legend: says "all for replicates" when it should be "four".

Corrected

- Lines 162-165: this is *extremely confusing*, and remains so even after looking at Supp Figure 1. I think what the authors are saying is that most sequencing reads correspond to shared

barcodes, but that not all detected barcodes are shared. This is expected if some barcodes are really very low frequency or are actually just sequencing errors. This point does also call into some question the relevance of the “unique barcodes” metric in Fig 3A if many of the barcodes are actually really low frequency or are sequencing errors. Accordingly, Fig 3 should provide comparable data to Fig 2 on the evenness of the barcodes before and after passaging.

We apologize for the lack of clarity. We have rewritten this section on lines 172-175. We have also included similar visualization and analyses of adapted stock before and after passaging in Fig 3 to allow comparison to Fig 2.

- It's not clearly explained if Figure 3D is showing pre- or post-passaging data for the new libraries. Also, what is the deal with the mutation near the C-terminus of HA? Is this another putative adaptive mutation?

We now clarify in the legend that Fig 3E (previously Fig 3D) shows the amplified (i.e. post-passage) viral stocks. We thank the reviewer for their observation about other SNVs within HA. We used this as an opportunity to re-test our variant calling. In the initial submission, the whole genome sequencing (WGS) data was analyzed using a custom bioinformatics pipeline previously used in the Friedrich lab. We found minor labeling errors that were reflected in the way the data were plotted before. In this re-submission, we have addressed that issue, and in response to the reviewer's comment, developed an updated pipeline to process all of our WGS data uniformly and under more refined parameters. We now use bbnorm to downsample our data to a target of 2,000 reads, LoFreq to call variants with both quality and read parameter thresholds (see updated methods), have filtered out variants below the 3% frequency threshold set for confidently calling SNVs versus sequencing error, and plotted both synonymous and non-synonymous variants (with labels for sub consensus variants such as K153E). The updated analysis pipeline can be found on lines 509-519. New data and plots are used in Fig 3E and 6C that more accurately represent SNVs in our samples. Note that apparent synonymous changes aberrantly called by our prior pipeline are no longer detected in the data.

- Figure 4C, I don't like the “total” barcode metric. Due to the various issues about barcode skewing, sequencing errors, etc I expect this metric will be susceptible to things like different sequencing depths and isn't a good indicator. How about using something like the inverse Simpson index, which calculates the effective diversity in a way that isn't susceptible to skewing, sequencing errors, depth, etc. In fact, I think the inverse Simpson index would be a more intuitive measure than the Shannon diversity in all cases as it can be directly interpreted as a measure of effective number of unique barcodes. The Shannon diversity doesn't have a natural numerical interpretation and depends on the log base you choose.

We appreciate this suggestion and have incorporated both Shannon's and the Gini-Simpson index into our analyses (new Fig 2B, 3A-B, 4D, 5D). We state the rationale for this approach on lines 130-137. New methods are added for this in the section beginning on line 565.

- Line 217: related to a point I made above, I don't think that's very surprising. I think there is quite a bit of evidence that the effective rate of intra-host reassortment isn't all that high.

Prior work by Marshall, et al. 2013 (23785286) showed high degrees of reassortment between matched viruses, both in culture and in animals. Reassortment frequency increased with MOI. While our low MOI infections in culture might produce limited co-infection and reassortment, our

animal models used high infectious doses that should favor reassortment. Marshal, et al. report on average ~60% of isolates from high-dosed animals were reassortants. A recent preprint from the Lowen lab has also demonstrated extensive reassortment in guinea pigs, ferrets and swine, although consistent with our findings, they see strong evidence for compartmentalization with the ferret respiratory tract (Ganti, et al. 2022 doi:10.1101/2022.02.08.479600). Given that viruses within our libraries are identical except for their barcodes, prior work led us to expect frequent reassortment, hence our surprise at data suggesting the opposite.

- I actually found Supp Figures 4 and 5 quite a bit more intuitive than much of what is plotted in Figure 6 and especially Figure 5 for showing the differences in variants among anatomical sites. The current presentation is adequate, but do consider adding some simpler more direct plots to main figures. The difference between anatomical sites is the coolest part of the paper, so it's worth making it as clear

Following the reviewer's suggestion, we rearranged the figures and include direct plots in Fig 5, moving titer data from the original Fig 5 into the supplement.

Reviewer #2 (Remarks to the Author):

Amato and colleagues added short barcodes to two influenza A virus genome segments, and went to great lengths to reconstitute virus populations with high levels of neutral barcode variation. These populations were then inoculated into the upper respiratory tract of both mice and ferrets, and population diversity was monitored as the virus was transmitted to the trachea and lungs. In mice, there was not clear evidence of population bottlenecks, whereas in ferrets there was clear evidence for population bottlenecks as the virus invaded the trachea and different lung lobes. The authors conclude that narrow population bottlenecks are an important feature of the within-host infection dynamics in the clinically relevant host (ferrets), and that these strong random effects may help to explain why there appears to be limited adaptive evolution within hosts.

This is an excellent and timely study on the within-host bottlenecks in virus populations, and an experimental tour de force. The authors have gone to great lengths to set up the system carefully, and also to provide a balanced and transparent presentation of the data and results. Below we have some questions about certain aspects of the study and comments on the presentation, which we hope the authors will find useful for improving the manuscript.

We thank the reviewers for their close reading of the manuscript and helpful suggestions.

Main comments:

The way the barcoding approach was implemented is excellent from a technical perspective. Repeating the barcoding for the preadapted variant was a simple but effective solution. One potential downside to this approach is the "background" signal. The authors have clearly taken some steps to mitigate the impact of sequencing errors (i.e., lines 473-481 in the methods section), but there is no validation of this approach and it does not guarantee all barcode variants are bona fide. We don't think this comes as any surprise to anyone who has worked with high-throughput sequencing (HTS) and we think the presentation of the data is fair, but we don't feel the authors have fully considered the implications of sequencing error in the interpretation of the results.

For example, the data suggest that some variants detected are low-frequency false positives (i.e., Fig. S6, where the "lung only" variants must be (i) sequencing errors, (ii) de novo

mutations, or (iii) variants missed in the wash). The observed patterns in lung lobes, one or a few variants at high frequency (e.g., Fig. 5I MR and LR, particularly for PA), suggest that one barcode was fixed here and the remaining signal is largely background. Do the authors agree? Is it possible to distinguish between the explanation suggested by the authors for these patterns (lines 294-296) and sequencing errors that remain unidentified? Because these issues are inherent to HTS and HTS is by far the best approach for characterizing viral populations, this issue is unavoidable unless CirSeq is used. But we do feel it can be better taken into account in the interpretation and presentation of the results.

We agree with the reviewers' point; the ability to uniquely identify variants, and the impact sequencing error has on this process, is a paramount concern for the accurate implementation of our approach. We have taken multiple steps to minimize process error, following best practices such as those in McCrone and Lauring 2016 (Pubmed 27194763). Nonetheless, errors will still occur. Therefore, we exploited the fact that theoretical barcode diversity was almost 10 times greater than our actual library size. This suggests that most "true" barcode lineages in our stocks are expected to differ by more than a single mutation. This allowed us to use clustering to consolidate read counts to consensus clusters and minimize the contribution of sequencing error while still counting those reads (Smith, 2017 28100584; UMI-Tools v1.1.1). Clustering identifies barcode networks linked by a single edit distance. The *adjacency* method used on our Illumina data takes into account the relative frequency of connected nodes to collapse subnetworks into consensus parental barcodes. The frequency of an error-derived barcode produced during sequencing is predicted to be much lower than its parental template that may already be present in the sample multiple times. If two closely related barcodes are present with similar frequency, they will be considered "true" barcodes and will not be consolidated, explaining why some of our processed data still contain highly similar barcodes. Given the higher error rate associated with long-read ONT sequencing, clustering of ONT data used the more aggressive *directional* approach that consolidates some of the unique subnetwork clusters that may have remained from the adjacency method. All reads within a network are then assigned to the parental barcode prior to quantitation and analysis. We mention this on lines 125-128. This does not fully eliminate the influence of sequencing errors, but affords a more conservative interpretation.

Even when we minimized errors, we often detected lineages in the lung that were not detected in the source of these infections, the nasal wash. This is most likely due to incomplete sampling. Our populations have a large number of low-frequency variants. Rarefaction-extrapolation analysis of data from the nasal wash of Ferret 34 suggests our sequencing detects only ~95% of the lineages present in the "true" population (Rebuttal Fig 1A-B). This was also observed in results from replicate sequencing runs of viral stocks, where rarefaction showed incomplete sampling and most barcodes found in single replicates were very low frequency (Supp Fig 1D-E). Line 177-183 makes clear that incomplete sampling may not detect some low frequency variants. For these reasons, we favor the hypothesis that the majority of "lung only" variants are distinct viral lineages, not sequencing artifacts, and their "lung only" phenotype arises from incomplete sampling of the nasal wash.

Rebuttal Fig 1. Very rare lineages contribute to incomplete sampling of viral population. A) Rarefaction-extrapolation and B) sample coverage lineages detected in the 3dpi nasal wash for ferret 34 suggest incomplete sampling due to very low frequency lineages.

The proposed explanation (lines 294-296) for the main mechanisms that underlie the observed distribution of genotypes is intuitive and we fully agree it is highly relevant. Although this explanation is pivotal for understanding these patterns, it is a pity the presentation doesn't come full circle by showing quantitatively that this explanation works. Ideally, one would estimate the dispersion in "takeoff" timepoints that best accounts for the data, but the authors would probably counter that this is beyond the scope of this presentation (and we would tend to agree). As a simple solution and presentation tool, therefore we suggest that the authors add a "seeding time" component to the simulations presented in Figure S5, to show that this mechanism can - in principle - lead to patterns that are more congruent with the data. We also think it is also important to provide some details about the model described here: the approach appears to be very straightforward, but it would be good to describe it explicitly.

We thank the reviewer for their suggestion. Simulated data modeled on our bottleneck estimates poorly matched the frequencies we detected in the lungs of ferrets. Our original submission proposed that this might be due to repeated seeding through the bottleneck. Following the reviewers comment, we introduced a variable time component and modeled outgrowth. However, these simulations did not perform better. Instead, we considered a single event when viruses pass through a bottleneck event to get to the lung lobe, but that additional environmental factors dictate the probability that any lineage will replicate efficiently. This would account for the possibility that some sites in the lung lobe are ideal for replication, whereas others are poorly suited. When this was modeled, the simulated data better reflected the observed data. This leads us to conclude that even though many lineages appear to seed the lung lobes, that once in the lobes, replication of viral lineages is subject to large stochastic effects, reducing viral diversity. These new results are discussed on lines 285-302 and incorporated into a new Sup Fig 5.

Minor comments:

Line 27: "Bottlenecks stochastically sampled.." should be "Bottleneck events stochastically sampled.."

Corrected.

Lines 29-31, 73-78 and elsewhere: Random events (i.e., small mutation supplies and population bottlenecks) can also be advantageous to evolution, when sign epistasis is common and populations can become trapped on suboptimal peaks (E.g., PNAS 112(24): 7530-7535, PNAS 114(48): 12773-12778). In fact, one could argue that the infection dynamics described here could be quite beneficial for long-term evolvability and effectively reconnoitering a rugged fitness landscape. Therefore, for a complete and balanced interpretation of the results, I would also consider this perspective on viral evolution. We realize the authors may not agree, but then we would argue why they feel this scenario is less relevant.

This is an excellent point and we have included it to balance our discussion. Lines 346-349.

Lines 33-41: Paradoxical. So at what level does selection then act? Primarily on the between-hosts level? They need to explain this further for a broad audience.

We make clear on line 39 that we do not currently know where selection acts.

Lines 43-57: At the authors' discretion: having an entire paragraph on H5N1 adaptation to mammals suggests that this work would also involve this strain or the adaptive process. It might be helpful to point out right away this is just an example to highlight the importance of evolutionary constraints to adaptation?

Done.

Line 48: The mutations that impede evolution/block trajectories need not be "deleterious" as such. They also can be beneficial or neutral, what matters is epistatic interactions (i.e., to be incompatible with other beneficial mutations).

Corrected.

Lines 60-62: High cellular MOI can also constrain evolution if the benefits (or costs) of mutations act in trans (E.g., J Virol 84(4):1828-37), by disassociating the cost/benefit from the genome precluding within-cell selection. Although there are evolutionary advantages to high MOIs, clearly there are also potential disadvantages. Useful to mention this here to make the argument more balanced.

This is an interesting point referencing very strict bottleneck events in wheat viruses. However, we prefer to keep the introduction focused.

Lines 96-97: Mention explicitly that the "result of selection" would be hitchhiking of barcoded variants with beneficial mutations?

Clarified on line 99.

Line 136: What was the effective population size here? Is it possible to provide an expectation a priori?

As noted above for Reviewer 1, we used 10^6 infectious virions spread across the 20 plates at an MOI of ~ 0.01 , and now on line 139.

Figures 2/3: Very informative figures, but it would be cool to display the diversity in the labelled variants made after preadaptation as is done in Fig. 2b. The contrast would be striking and it would make the results very intuitive (whilst we realize that Fig. 3 is technically more informative).

We have created the same visualization and statistical analyses, now present it in Fig 3.

Panels 5 f/g: It took a while to figure out that these are the tracheal samples. Make this clearer in the legend, and/or even better, add a visual cue in the figure itself?

We apologize for the confusion. Axes in Fig 5F-G are now labeled "trachea" to aid the reader.

Conceptually, since the authors anticipated tight bottlenecks, one could question whether this barcoding approach was needed. I.e., Could experimental results with a few markers (I.e., mixtures of NheI and PstI) have rendered the same result (i.e., maintenance of both variants with minimal changes in frequency nasally, followed by the consistent fixation of one variant in the trachea or lungs [for ferrets])? We would find this an interesting discussion point, and others contemplating similar work could greatly benefit from this.

Influenza virus libraries with smaller barcode populations have yielded key insights into viral transmission between hosts (e.g. Varble, et al. 2014 25456074). While we might have had initial predictions, our approach was agnostic to bottleneck size and the large barcoded populations were needed to accommodate all potential outcomes. This was apparent in the mouse model, and in ferret nasal washes, where populations remained very diverse. The scale of diversity and the lack of bottleneck events could not be as easily appreciated with a limited number of barcodes. Moreover, the large viral libraries afforded a high degree of resolution when selection occurred. The processes leading to tissue culture adaption and sweeps may not have been fully captured with a small viral library.

Line 454: Following the link leads to a “Link not found, 404 Error” message. Should this be the same link as line 438?

Fixed.

Supplemental Figure 1B: change “for” to “four”.

Done

Reviewer comments, second round of review

Reviewer #1 (Remarks to the Author):

I support publication of this very nice paper in the current form.

Reviewer #2 (Remarks to the Author):

Amato and colleagues have thoroughly revised their manuscript on within-host population bottlenecks during IAV spread in two different hosts. I would like to recognize the effort they have made in developing models that help account for the observed patterns. It is very interesting to see that the original explanation proposed (different times of seeding) is not supported, but rather a model that assumes heterogeneity in replication potential of the invaded patches.

At the authors' discretion: the model they suggested in the original manuscript (different times of seeding) is no longer mentioned at all, whereas it was considered and rejected. I think this was a reasonable suggestion, so why not briefly describe the predictions and rationale for rejecting this model in the manuscript?

Line 624: "Here, we let $\sigma = 2$." I understand that they did not parametrize the model, but chose a value for this free model parameter that leads to predicted patterns qualitatively similar to the data. I would suggest stating this explicitly, rather than having this essential model parameter appear out of thin air. Could the authors also briefly comment on the implications of this value, because it suggests an extreme range in the replication potential of patches in the lung (i.e., > 700 fold difference between 5% and 95% percentiles). Is there a plausible biological explanation for these differences and is it congruent with what is known about IAV replication in the lungs?

At the authors' discretion: I would comment on this modeling result in more detail in the discussion, otherwise it is likely to be lost on many readers. I think the picture this model sketches of invasion and replication dynamics in the lower airways is very interesting, suggesting that it is the replication kinetics in small patches that dictate the patterns seen at a sub-organ level.

REVIEWER COMMENTS

We thank the reviewers for their re-reading of our submission and positive assessment. Point-by-point responses to the requested text changes are addressed below.

Reviewer #1 (Remarks to the Author):

I support publication of this very nice paper in the current form.

Thank you for this positive assessment and support for publication.

Reviewer #2 (Remarks to the Author):

Amato and colleagues have thoroughly revised their manuscript on within-host population bottlenecks during IAV spread in two different hosts. I would like to recognize the effort they have made in developing models that help account for the observed patterns. It is very interesting to see that the original explanation proposed (different times of seeding) is not supported, but rather a model that assumes heterogeneity in replication potential of the invaded patches.

At the authors' discretion: the model they suggested in the original manuscript (different times of seeding) is no longer mentioned at all, whereas it was considered and rejected. I think this was a reasonable suggestion, so why not briefly describe the predictions and rationale for rejecting this model in the manuscript?

We thank the reviewer for this suggestion. There are many different ways in which seeding at different times could be implemented in model simulations. For example, we could imagine a scenario in which the viral population in the nasal compartment grows exponentially, and seeding occurs at a per-virion rate, such that the amount of seeding - quantified by the total number of virions being transmitted to another compartment per unit time - increases over time. Alternatively, we could imagine a scenario where, although the viral population in the nasal compartment might be growing exponentially, the amount of seeding may be constant over time because the recipient tissue becomes less 'permissible' for viral replication, perhaps because of a local interferon response. Because there are many possibilities that can be implemented, we do not feel confident in concluding that seeding at different time points is not the underlying dynamical process that gives rise to the relatively weak correlation of lineage frequencies between nasal wash samples and lung lobe samples. We have added the following text to the Discussion section of the manuscript to raise the possibility that other processes (besides local stochastic replication dynamics) could perhaps result in a similar pattern, or that a combination of processes might be at play:

Line 337: Our results show that influenza virus evolution within a host involves multiple stochastic or noisy processes, tempering the impact of positive selection and likely creating additional barriers for host-adaptation and onward transmission. This could involve multiple physical bottlenecks as the virus sequentially infects distinct tissues and stochastic replication dynamics within a focal tissue following its infection. Our modelling provides evidence that both of these processes may be at play, and suggests that replication kinetics in these largely isolated populations may play a critical role in shaping patterns of viral diversity detected at a sub-organ level. While the model simulations with heterogeneous viral replication were able to recapitulate observed patterns of viral diversity and the relatively weak correlations between nasal wash and lung lobe lineage frequencies, we cannot definitively conclude that

heterogeneous viral replication is the process that drives these patterns. Alternative processes not considered explicitly here may also be able to reproduce similar patterns. For example, repeated seeding from the upper respiratory tract may allow viral lineages that reach the lower respiratory tract earlier to expand before the arrival of other viral lineages. This would have a similar effect of reducing evenness and diversity of viral populations in the lung lobes relative to those sampled in nasal washes. We would also expect a process of repeated seeding to yield weak correlations between nasal wash and lung lobe frequencies. Of course, heterogeneous viral replication within lobes and repeated seeding are also not mutually exclusive processes, and both may be at play in shaping patterns of viral diversity in the lower respiratory tract. Future work is needed to be able to experimentally assess the relative roles of these two processes, and potentially others, in driving these patterns of viral diversity.

Line 624: "Here, we let $\sigma = 2$." I understand that they did not parametrize the model, but chose a value for this free model parameter that leads to predicted patterns qualitatively similar to the data. I would suggest stating this explicitly, rather than having this essential model parameter appear out of thin air. Could the authors also briefly comment on the implications of this value, because it suggests an extreme range in the replication potential of patches in the lung (i.e., > 700 fold difference between 5% and 95% percentiles). Is there a plausible biological explanation for these differences and is it congruent with what is known about IAV replication in the lungs?

Several reports revealed that the probability of successful infection varies greatly for influenza virus. Infection is a highly heterogenous process at the local scale. Culture-based studies revealed that upwards of 90% of infected cells fail to produce infectious virions (Brooke, et al. 2013 PMID 23283949). Single-cell analysis has revealed this is impacted by the high degree of heterogeneity both within the viral population and in cellular responses (Heldt, et al. 2015 26586423; Russell, et al. 2018 29451492; Russell, et al. 2019 31068418; Sun, Vera et al. 2020 32614923). In particular, Heldt, et al. showed that viral output between infected cells can span 3 orders of magnitude. *In-vivo* experiments also revealed heterogeneity in infection across different sites, contributing to compartmentalized replication (Ganti, et al. 2022, doi: 10.1101/2022.02.08.479600). Thus, models like ours that allow for this high degree of variability reflect biologically plausible processes.

At the authors' discretion: I would comment on this modeling result in more detail in the discussion, otherwise it is likely to be lost on many readers. I think the picture this model sketches of invasion and replication dynamics in the lower airways is very interesting, suggesting that it is the replication kinetics in small patches that dictate the patterns seen at a sub-organ level.

We have added the following text to the Methods section to describe our choice of σ in more detail:

line 553: We considered a range of σ values. With $\sigma = 0$, the original multinomial model described above is recovered, leading to strong correlations between viral lineage frequencies in the nasal wash and the lung lobes that are not empirically observed. At higher values of σ , model-projected evenness and diversity decreased, and the projected correlations between lineage frequencies in the nasal wash and the lung lobes became weaker. Here, we let $\sigma = 2$ for each of the lung lobes, based on simulation results that indicated that this level of replication heterogeneity could recover the weak (but apparent) observed correlation between nasal wash and lung lobe lineage frequencies.

We have further made the following edits to the Results section, highlighting that the simulations that recapitulated the weak lineage frequency correlations were parameterized with a substantial amount of lobe heterogeneity:

Simulations of this modified model, parameterized with a substantial amount of lobe heterogeneity (> 700 fold replication difference between the 5% and 95% percentiles of the viral population), yielded correlations more similar to those observed between nasal wash and lung lobe lineage frequencies.